# Comprehensive Evaluation of Nutritional Qualities of Chinese Cabbage (*Brassica rapa* ssp. *pekinensis*) Varieties Based on Multivariate Statistical Analysis

**Chao Song** [1,2,†], **Xinyu Ye** [1,2,†], **Guangyang Liu** [1], **Shifan Zhang** [1], **Guoliang Li** [1], **Hui Zhang** [1], **Fei Li** [1], **Rifei Sun** [1], **Chenggang Wang** [2], **Donghui Xu** [1,*] and **Shujiang Zhang** [1,*]

[1]  State Key Laboratory of Vegetable Biobreeding, Institute of Vegetables and Flowers, Chinese Academy of Agricultural Sciences, Beijing 100081, China; 17795115425@163.com (C.S.); yexinyu1012@126.com (X.Y.); liuguangyang@caas.cn (G.L.); zhangshifan@caas.cn (S.Z.); liguoliang@caas.cn (G.L.); zhanghui05@caas.cn (H.Z.); lifei@caas.cn (F.L.); sunrifei@caas.cn (R.S.)
[2]  College of Horticulture, Anhui Agricultural University, Hefei 230031, China; cgwang@ahau.edu.cn
[*]  Correspondence: xudonghui@caas.cn (D.X.); zhangshujiang@caas.cn (S.Z.)
[†]  These authors contributed equally to this work.

**Abstract:** In order to make the identification and utilization of nutritional quality components in Chinese cabbage more predictive, to obtain ideal raw materials, and to help screen functional Chinese cabbage varieties that have high nutritional value, we conducted a comprehensive evaluation of the nutritional quality of different Chinese cabbage varieties. In this study, 17 nutritional quality indexes of 35 Chinese cabbage varieties, including crude fiber (CF), crude protein (CP), vitamin C (VC), glucose (Glc), fructose (Fru), malic acid (MA), citric acid (CA), oxalic acid (OA), total amino acid (TAA) and CA, K, Mg, P, Cu, Fe, Mn and Zn, were analyzed using diversity analysis, correlation analysis, principal component analysis, membership function analysis and cluster analysis. The results showed that there were different degrees of variation in the 17 nutritional quality indexes, and the coefficients of variation ranged from 11.45% to 91.47%. The correlation analysis found that there were significant or extremely significant correlations between different nutrient elements of Chinese cabbage, which indicated that principal component analysis could be carried out, and the comprehensive score (D value) of different materials could be obtained using principal component analysis and the membership function method. The nutritional quality of Chinese cabbage was classified into five grades by cluster analysis. Finally, a mathematical model for evaluating the nutritional quality of Chinese cabbage was established using the D value and multiple stepwise regression methods, and 10 key indexes were selected from the 17 indexes, which could be used for the rapid identification of the nutritional quality of Chinese cabbage. This study provided a theoretical basis for the nutritional quality evaluation and variety breeding of Chinese cabbage.

**Keywords:** Chinese cabbage; nutritional quality; PCA; membership function method; comprehensive evaluation

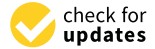



## 1. Introduction

Chinese cabbage (*Brassica rapa* L. ssp. *pekinensis*) is an economically important leaf vegetable of the brassica genus, growing worldwide, especially in Asian countries. The annual production of Chinese cabbage in Asia exceeds 50 megatons, accounting for about 70% of the world's production [1]. According to incomplete statistics, the annual planting area of Chinese cabbage is 2.67 million hm$^2$, accounting for about 13% of the total planting area of vegetables in the country, and its output value exceeds CNY 60 billion, making it the largest vegetable crop in China. With the continuous improvements in living standards and the increasing variety of vegetable supplies, people have higher requirements for the quality of vegetables, including Chinese cabbage. Therefore, the selection, improvement

and optimization of Chinese cabbage varieties are urgent, and improving their nutritional quality is already an important goal for Chinese cabbage breeding [2].

The quality of Chinese cabbage includes commodity quality, flavor quality and nutritional quality [3]. Commodity quality is external quality, and includes freshness, leaf ball tightness, etc., which are mainly determined by the senses; flavor quality refers to the unique smell and taste of Chinese cabbage, which need to be identified by tasting and analyzing the flavor-related components. The nutritional quality is determined by the content and distribution ratio of nutrients, such as crude protein, soluble sugar, fiber, organic acid, amino acid and mineral elements, and is related to human health. At present, the quality of Chinese cabbage at home and abroad mainly focuses on the evaluation of sensory and flavor quality [4–6]. However, in the reports on the nutritional quality components of Chinese cabbage, the cost may be very high due to the complex determination methods of nutritional quality components and the high costs of instruments, equipment and reagents, which may lead to an insufficient selection of varieties. Due to the limited nutritional indexes [3,7], the quality of a Chinese cabbage cannot be comprehensively and accurately judged.

A single nutrient component index cannot complete a comprehensive evaluation of Chinese cabbage, and the nutritional quality of different varieties of Chinese cabbage can be more comprehensively evaluated based on the analysis of multiple nutrient component indexes of different varieties [8]. Multivariate statistical methods have played an important role in which quality assessments are made against a background of multiple food attributes. Multivariate analysis effectively describes the relationship between the variables of interest in a study and is applicable when multiple measurements are taken on different individuals or objects in one or more samples [9]. At present, the evaluation model has been developed from a simple analytic hierarchical process [10,11] and entropy weight method [12] to multi-attribute comprehensive evaluation, such as principal component analysis [13,14], grey correlation analysis, membership function analysis and TOPSIS analysis [15–17]. However, from the comparison of evaluation models, the obvious diversity of evaluation results restricts objective decision making and scientific field management due to differences in the subjective behavior of evaluators, the selectivity of the method structure and the discarding of evaluation information [18].

In order to improve the breeding efficiency of Chinese cabbage varieties, a comprehensive evaluation method of the nutritional quality of Chinese cabbage is urgently needed. Principal component analysis (PCA), an unsupervised pattern recognition technique, is an important tool for visualizing similarities or dissimilarities in multivariate data [19,20]. In addition, the membership function method eliminates the effect of excessive extreme values on certain indicators, and scales the data between 0 and 1 to achieve normalization; it has been applied to evaluate ornamental and resistance traits in chrysanthemum [21]. At the same time, an evaluation method combining principal component analysis and membership function has been applied to the comprehensive evaluation of rice quality [22]. Hierarchical cluster analysis (HCA) determines the similarity between samples by measuring the distance between all possible sample pairs in a high-dimensional space, and all the similarities between samples are represented by a two-dimensional graph [23]. Hierarchical Clustering Dendrogram allows the visualization of datasets and is also considered a useful germplasm assessment technique. Combining the results of PCA and HCA makes it easy and intuitive to understand the differences between samples and to find good hybrids of interest [24,25]. Multiple linear regression (MLR) uses a step-by-step algorithm to predict the outcome of a response variable with multiple explanatory variables. MLR can not only be used to establish the relationship between spectral VIs and the crop characteristics studied but also to select the most informative variables to estimate crop characteristics [26]. Previously published studies have reported that MLR is a widely used method to rapidly estimate leaf crop nitrogen concentrations [27,28] and grain yields [29,30], whereas PCA, HCA and MLR are also used as important indicators to evaluate lychee browning and rot [31].

Therefore, in this study, hierarchical cluster analysis, principal component analysis, membership function method, and multiple stepwise regression analysis were used to comprehensively evaluate the nutritional quality of 35 Chinese cabbage varieties, establish an evaluation model, determine the important indexes for the nutritional quality of Chinese cabbage, and screen out the most nutrient-rich Chinese cabbage varieties under conventional cultivation conditions, in order to provide some guidance for the improvement and breeding of nutritious Chinese cabbage.

## 2. Materials and Methods

### 2.1. Plant Materials

The 35 fresh Chinese cabbage varieties collected in this experiment were harvested on 16 September 2020 in Shunyi, Beijing, China (40°13′ N, 116°65′ E); high ridge cultivation was adopted in the open field. These varieties included virtually all types of Chinese cabbage on the market. The individual names, provenance details and agronomic traits of these 35 cultivars are described in Appendix A (Table A1). In addition to genotypic differences, these Chinese cabbages grown on the same plot had consistent cultivation conditions and daily management practices, plots were arranged in a randomized block pattern with three replicates, the planting area was 7 m$^2$ (border length 7 m, width 1 m), 2 rows were planted in each plot, 15 plants in each row, the row spacing was 0.5 m, and the plant spacing was 0.5 m. Three kinds of Chinese cabbage with the same size, similar maturity and no pests and diseases were selected from each variety for the experiment. In order to prevent the interference of external impurities, the outer leaves of rolled Chinese cabbage were peeled off; the percentage of removed leaves in the weight of cabbage varied depending on the variety, ranging from 27% to 50%, and the leaf balls were sampled by a four-fold method, chopped and mixed, and repeated three times.

### 2.2. Sample Preparation and Determination of Nutritional Constituents

Each finely mixed sample was divided into three parts, and 100 g of freshly chopped sample was weighed in one part for the determination of vitamin C; in the second part, a sufficient amount of minced samples were weighed and homogenized with a wall breaker for the determination of organic acids, sugar components and free amino acids; in the third part, sufficient samples were taken in a tray and dried at 65 °C to constant weight, ground with a grinding mill after drying, sealed in a zipper bag and stored in the dryer until use. Fresh materials were collected and tested on the experimental platform of the Vegetable Quality Supervision and Inspection Testing Center of the Ministry of Agriculture and Rural Affairs. The methods and main instruments for measuring different nutritional quality contents are shown in Appendix A (Tables A2 and A3).

According to the national standard GB 5009.10-2003, the determination of crude fiber in plant-based foods was carried out [32]. According to the national standard GB 5009.5-2016 for the determination of protein in food, the combustion method was used to determine the crude protein [33]. According to the national food safety standard GB 5009.268-2016, inductively coupled plasma atomic emission spectrometry (ICP-OES) was used to determine the mineral elements in the food [34]. The agricultural standard NYT2277-2012 was referred to in the determination of organic acids and anions in fruits and vegetables—ion chromatography to determine the organic acids content [35]. Referring to the national food safety standard GB 5009.86-2016 for the determination of ascorbic acid in food, the 2,6-dichloro indophenol titration method was used to determine the vitamin C content [36]. Referring to the national standard "GB/T 30987-2020" for the determination of free amino acids in plants, the fully automatic amino acid analyzer method was used to determine the free amino acids content. Due to the low content of free amino acids in Chinese cabbage, the sample weight used in this study was 10 times that of the standard sample [37]. The determination of fructose, glucose, sucrose, maltose and lactose in the food was carried out with reference to national standards for food safety "GB 5009.8-2023". The determination

of sugar components was carried out via ion chromatography [38]. Detailed procedures for all determination methods are provided in Appendix A.

### 2.3. Multivariate Statistical Analysis

Microsoft Excel 2016 software was used to process the data. The SPSS 22.0 software (SPSS, Inc., Chicago, IL, USA) was used for descriptive statistics, correlation analysis, principal component analysis (PCA) and cluster analysis, and the website (https://www.genescloud.cn/cloudClassroom, accessed on 10 September 2023) was accessed on 10 September 2023 and used to create the figures; all significant differences were conducted at $p < 0.05$.

The principal components of quality components were extracted for PCA, and the initial eigenvalues, variance contribution rates and cumulative contribution rates of principal components were obtained. A fuzzy mathematics membership function method was used to obtain the membership function value of each comprehensive index. The main quality components were analyzed and evaluated using cluster analysis. The weight value ($W_i$) of the main quality components was calculated from the variance contribution rate, and then the weighted score value (D value) of the quality was calculated according to the membership function method and the calculated formulas were obtained. Finally, multiple stepwise regression equations were established using the D value, and the key indexes for judging the nutritional quality of Chinese cabbage were obtained.

Membership function values of each quality component index of different Chinese cabbage materials were established as follows:

$$u(X_i) = (X_i - X_{min})/(X_{max} - X_{min}), i = 1, 2, 3, \cdots, n \tag{1}$$

In the formula, $u(X_i)$ is the membership function value; $X_i$ represents the $i$th component index; $X_{max}$ represents the maximum of the i-component index; $X_{min}$ represents the minimum value of the i-component index.

The weight of each nutrient index is determined as follows:

$$w_i = P_i / \sum_{i=1}^{n} p_i \tag{2}$$

In the formula, $w_i$ represents the importance of the $i$th component index in all component indexes, that is, the weight; $p_i$ represents the contribution rate of the first component index of Chinese cabbage obtained via the principal component analysis.

$$D = \sum_{i=1}^{n} [u(X_i) \times w_i] \tag{3}$$

In the formula, the D value is the comprehensive evaluation value for the quality of Chinese cabbage varieties.

## 3. Results

### 3.1. Difference Analysis of Nutritional Constituents in 35 Varieties of Chinese Cabbage

3.1.1. Component Content Analysis

Figure 1 shows that there were significant differences in the distribution of nutrients among different Chinese cabbage varieties. The specific content of the 17 different nutritional indexes of 35 kinds of Chinese cabbage is shown in Appendix A. The results indicated the content at fresh weight (Tables A4–A6). In terms of CP, Glc, Fru, MA, CA, OA, VC, TAA, Ca, K, Mg, P, Cu, Fe, Mn and Zn, many varieties ranked in the top three for multiple nutrient content, such as 20, 33 and 34 (Figure 1). In terms of CF, the three Chinese cabbage varieties with the least abundant content were 18, 9 and 12. These results suggest that the top three varieties rich in free amino acids and other nutrients may have better flavor and nutritional value, whereas the top three varieties with the least crude fiber may have better taste.

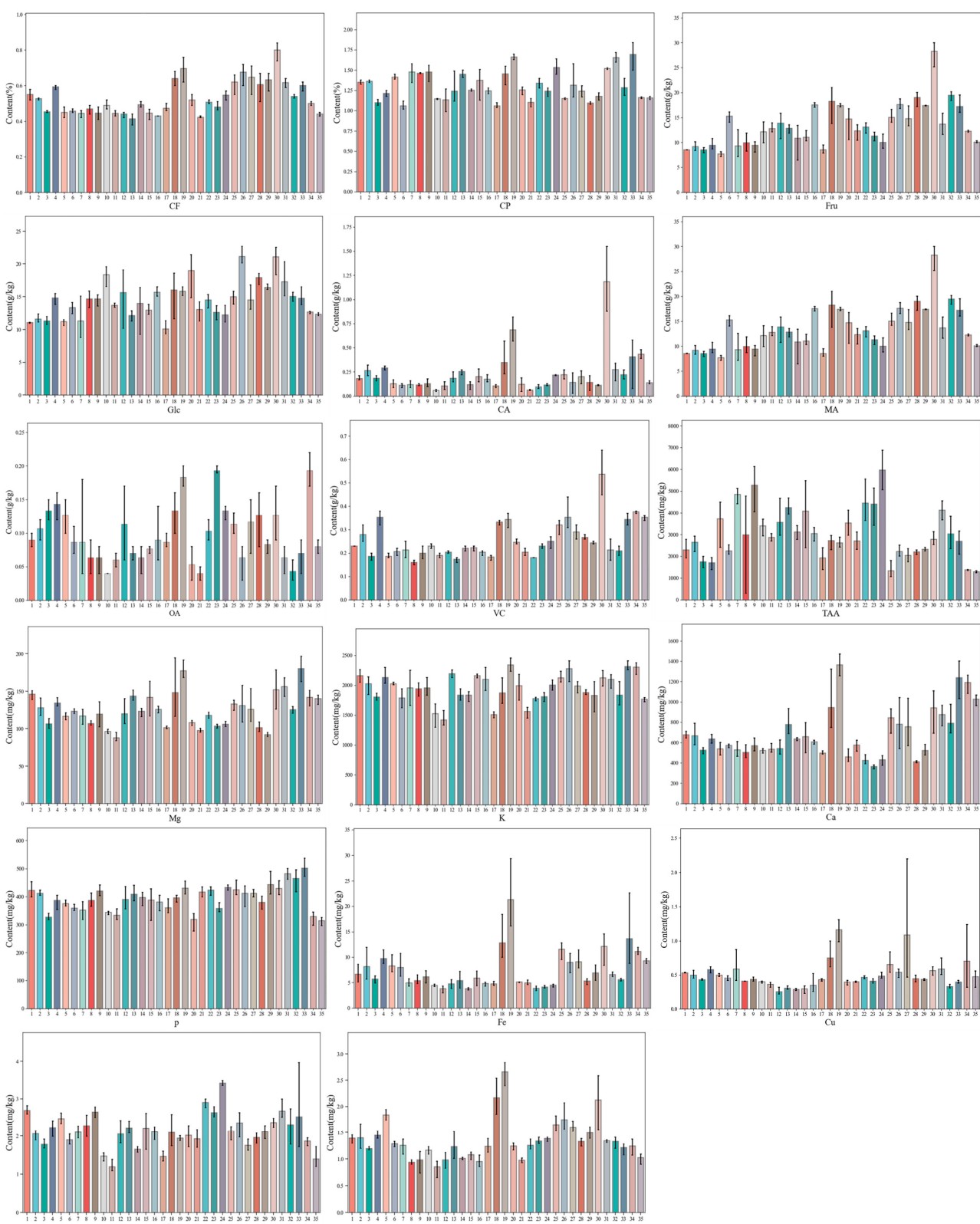

**Figure 1.** Determination of 17 kinds of nutrients in 35 Chinese cabbage varieties.

### 3.1.2. Diversity Analysis

The diversity of different nutrients in the Chinese cabbage varieties fully reflects the diversity of their genetic backgrounds, as the 35 Chinese cabbage cultivars in this study were consistent in terms of site conditions, cultivation and management conditions. The

results showed that the two highest contents of the 17 nutrients in the 35 varieties of Chinese cabbage were Glc and Fru, which reached 14.52 and 13.40 g/kg (Table 1). Compared to previous studies, it was found that soluble sugar is indeed one of several nutritional qualities with a higher content, which is consistent with the results of our study [39,40]. Coefficient of variation (CV) of less than 10% was used as the measure of a small variation degree, 10−20%, as a medium variation degree, and more than 20% as a high variation degree [41]. In this study, the coefficient of variation of each nutrient composition ranged from 11.45% to 91.47%, and there were no nutrients with a small variation degree. Except for CP, CF, Glc, K, P and Fe, the other nutrients were all high variation degree (Table 1). According to the above criteria, it can be seen that different nutrients are highly variable, especially the coefficient of variation of CA reached 91.47%, which means that each nutrient for the 35 Chinese cabbage varieties has a high utilization space and the potential for genetic improvement.

**Table 1.** Estimates of descriptive statistics (including the min, max, mean, SD, SE and CV) for the mineral element (mg/kg) and other nutrient (g/kg) contents of 35 Chinese cabbage varieties.

| Index | Range | Min | Max | Mean | SD | SE | CV (%) |
|---|---|---|---|---|---|---|---|
| CP | 6.30 | 10.60 | 16.90 | 13.13 | 1.78 | 0.30 | 13.53 |
| CF | 3.80 | 4.20 | 8.00 | 5.30 | 0.95 | 0.16 | 17.86 |
| Glc | 11.05 | 10.09 | 21.14 | 14.52 | 2.73 | 0.46 | 18.83 |
| Fru | 20.54 | 7.72 | 28.26 | 13.40 | 4.30 | 0.73 | 32.08 |
| MA | 2.21 | 0.22 | 2.43 | 0.84 | 0.46 | 0.08 | 54.34 |
| CA | 1.12 | 0.06 | 1.18 | 0.23 | 0.21 | 0.03 | 91.47 |
| OA | 0.16 | 0.04 | 0.20 | 0.10 | 0.04 | 0.01 | 41.97 |
| VC | 0.38 | 0.16 | 0.54 | 0.25 | 0.08 | 0.01 | 31.26 |
| TAA | 4.60 | 1.29 | 5.89 | 3.08 | 1.21 | 0.20 | 39.25 |
| Ca | 998.28 | 366.33 | 1364.61 | 684.26 | 243.29 | 41.12 | 35.56 |
| K | 922.4 | 1420.33 | 2342.73 | 1952.19 | 45.24 | 39.29 | 11.45 |
| Mg | 92.75 | 87.71 | 180.46 | 125.04 | 0.19 | 3.78 | 39.08 |
| P | 188.20 | 313.94 | 502.14 | 394.98 | 232.44 | 0.11 | 11.91 |
| Cu | 0.91 | 0.25 | 1.16 | 0.50 | 3.68 | 0.03 | 49.83 |
| Fe | 17.52 | 3.82 | 21.34 | 7.38 | 22.36 | 0.62 | 17.89 |
| Mn | 1.81 | 0.85 | 2.66 | 1.35 | 0.38 | 0.06 | 28.38 |
| Zn | 2.22 | 1.20 | 3.42 | 2.14 | 0.45 | 0.08 | 21.02 |

CP, crude protein; CF, crude fiber; Glc, glucose; Fru, fructose; MA, malic acid; OA, oxalic acid; CA, citric acid; VC, vitamin C; TAA, total amino acid; index, different indicators of nutritional quality; range, difference between maximum and minimum; Min, minimum; Max, maximum; mean, mean of all samples; SD, standard deviation; SE, standard error; CV, coefficient of variation.

### 3.1.3. Correlation Analysis

Some studies have shown that there is a certain internal relationship between different nutrients. Pearson correlation analysis can reveal the degree of correlation between two parameters. After correlation analysis of the data obtained, we also found that there were significant or extremely significant ($p < 0.05$ or $p < 0.01$) correlations between different nutritional elements of Chinese cabbage in many groups (Figure 2). With either significant positive correlation or significant negative correlation, the correlations between Ca and Mg (r = 0.857, $p < 0.01$), Ca and Fe (r = 0.847, $p < 0.01$), CA and VC (r = 0.780, $p < 0.01$), and TAA and VC (r = −0.445, $p < 0.01$) were higher. These results suggest that there are intrinsic relationships among these nutrients, leading to overlapping information, and further suggesting that principal component analysis can be performed on the basis of analyzing the internal relationships between components.

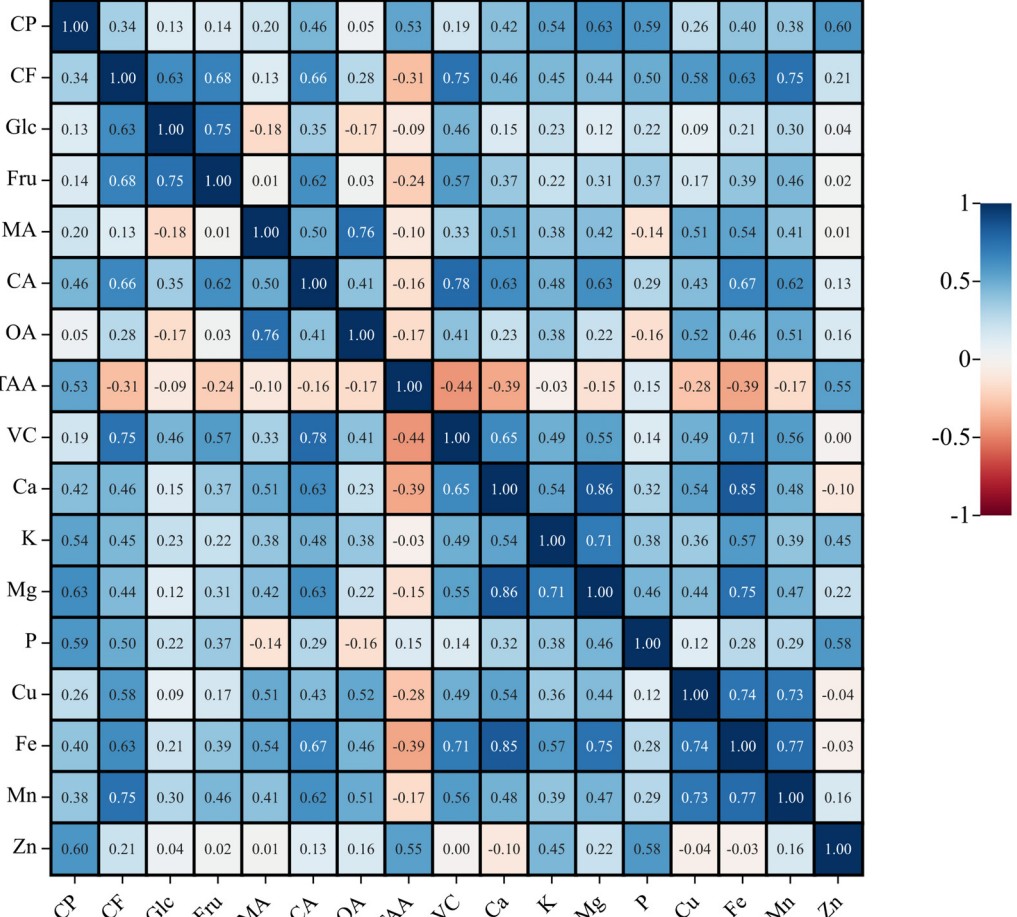

**Figure 2.** Correlation matrix based on Pearson's correlation coefficients between different nutrient contents. The intensity and number of colors are directly proportional to the correlation coefficients.

### 3.1.4. Principal Component Analysis

Principal component analysis (PCA) was used to analyze 17 quality components of 35 Chinese cabbage varieties. The number of extracted principal component factors was determined according to the principle that the eigenvalue is greater than 1 and the steep slope becomes gentle. In this study, although the eigenvalue of the fourth principal component was slightly greater than 1, the data of the fourth principal component and its subsequent eigenvalues tended to be flat, so we judged that the first three principal component factors could be extracted. The results show that the variance contribution rate of the first principal component (PC1) is 43.27%, the second principal component (PC2) is 15.96%, the third principal component (PC3) is 13.48%, and the cumulative contribution rate of the first three characteristic roots is 72.70%, which can represent most of the information of the original data (Table 2).

Among the feature vectors of PC1, the coefficients of CP, CA, VC, Ca, Mg, Fe and Mn are relatively high, indicating that the score of PC1 is determined by these components. Similarly, among the feature vectors of PC2, the coefficients of TAA, P, Cu and Zn were relatively high, indicating that the score of PC2 was determined by these components. Among the feature vectors of PC3, the coefficients of CF, Glc, Fru, MA, OA and K were relatively high, indicating that the score of PC3 was determined by these components.

**Table 2.** The eigenvectors and contribution rates of the principal components of each nutritional quality component.

|  | PC1 | PC2 | PC3 |
|---|---|---|---|
| CP | 0.072 | 0.260 | 0.116 |
| CF | 0.111 | 0.023 | 0.174 |
| Glc | 0.054 | 0.056 | 0.327 |
| Fru | 0.079 | 0.018 | 0.288 |
| MA | 0.072 | 0.120 | 0.270 |
| CA | 0.114 | 0.003 | 0.029 |
| OA | 0.066 | 0.130 | 0.231 |
| TAA | 0.038 | 0.274 | 0.133 |
| VC | 0.111 | 0.087 | 0.103 |
| Ca | 0.110 | 0.054 | 0.037 |
| K | 0.095 | 0.099 | 0.100 |
| Mg | 0.108 | 0.063 | 0.085 |
| P | 0.060 | 0.250 | 0.074 |
| Cu | 0.094 | 0.102 | 0.092 |
| Fe | 0.123 | 0.077 | 0.047 |
| Mn | 0.110 | 0.025 | 0.006 |
| Zn | 0.028 | 0.294 | 0.110 |
| Eigenvalue | 7.355 | 2.713 | 2.291 |
| Variance contribution rate | 43.266 | 15.960 | 13.478 |
| Cumulative contribution rate | 43.266 | 60.600 | 3.740 |

CP, crude protein; CF, crude fiber; Glc, glucose; Fru, fructose; MA, malic acid; OA, oxalic acid; CA, citric acid; VC, vitamin C; TAA, total amino acid; PC, principal component.

### 3.2. Comprehensive Evaluation of Nutritional Constituents of 35 Chinese Cabbage Varieties Based on PCA

#### 3.2.1. Membership Function Analysis and Comprehensive Score

SPSS 22.0 data processing software was used to standardize the data of each individual index, and the corresponding product of the standardized data and the index coefficients of each principal component comprehensive index obtained by the principal component analysis were carried out to calculate the values of three comprehensive indexes [CI(x)] of each Chinese cabbage variety. According to Formulas (1)–(4), the membership function value [$u(X_i)$] of three principal component comprehensive indexes of each material, the weight of each principal component comprehensive index ($w_i$) and the comprehensive score D value of each material were calculated, and the D value reflected the quality of nutrient components of each material (Table 3). The D values of the top-ranked varieties were higher, which indicated that they had a high comprehensive quality. Ranking the D value of each Chinese cabbage variety, the top five varieties are 19, 30, 33, 18 and 31.

#### 3.2.2. Cluster Analysis of Nutrient Composition of Different Chinese Cabbage Varieties

As shown in Figure 3, when the Euclidean distance is equal to 7.5, Chinese cabbage materials can be divided into five categories (Figure 3). Group I included two varieties of Chinese cabbage, including 30 and 19 (D ≥ 0.66). Group II included two varieties of Chinese cabbage, including 33 and 18 (0.49 ≤ D ≤ 0.54). Group III included six varieties of Chinese cabbage, including 34, 31, 4, 26, 27 and 25 (0.33 ≤ D ≤ 0.44). Group IV included 20 varieties of Chinese cabbage, including 29, 22, 23, 16, 35, 7, 12, 20, 28, 8, 14, 6, 13, 15, 5, 32, 9, 1, 24 and 2 (0.19 ≤ D ≤ 0.31) Group V included five varieties of Chinese cabbage, including 21, 10, 3, 17 and 11 (0.1 ≤ D ≤ 0.15). Group I is rich in various nutrients except TAA, P and Zn. The contents of TAA, P and Zn in group 2 were the highest (Table 4). Therefore, in our daily life, we only need to choose the cabbage from group 1 and group 2, which are basically able to meet our needs.

**Table 3.** The scores of three comprehensive indexes, the membership function value and the comprehensive score D value of each material.

| Variety | CI(1) | CI(2) | CI(3) | U(1) | U(2) | U(3) | D Value |
|---|---|---|---|---|---|---|---|
| 1 | 0.30 | 0.78 | 1.46 | 0.31 | 0.28 | 0.32 | 0.30 |
| 2 | 0.17 | −0.18 | 1.04 | 0.33 | 0.27 | 0.31 | 0.31 |
| 3 | −2.13 | −2.35 | 1.21 | 0.13 | 0.09 | 0.27 | 0.15 |
| 4 | 1.33 | −1.05 | 0.26 | 0.34 | 0.21 | 0.45 | 0.33 |
| 5 | −0.77 | 0.72 | 2.27 | 0.28 | 0.32 | 0.28 | 0.29 |
| 6 | −1.42 | −1.39 | −0.45 | 0.18 | 0.17 | 0.28 | 0.19 |
| 7 | −1.69 | 0.82 | 1.45 | 0.24 | 0.34 | 0.22 | 0.26 |
| 8 | −2.20 | 1.55 | 0.27 | 0.17 | 0.31 | 0.25 | 0.21 |
| 9 | −1.55 | 2.69 | 0.98 | 0.23 | 0.44 | 0.26 | 0.28 |
| 10 | −2.84 | −0.99 | −2.07 | 0.11 | 0.20 | 0.23 | 0.15 |
| 11 | −3.65 | −1.98 | −1.00 | 0.06 | 0.14 | 0.17 | 0.10 |
| 12 | −1.28 | 0.23 | 0.01 | 0.17 | 0.20 | 0.41 | 0.22 |
| 13 | −1.03 | 1.30 | 1.01 | 0.31 | 0.30 | 0.24 | 0.29 |
| 14 | −2.08 | −0.03 | −0.59 | 0.18 | 0.22 | 0.24 | 0.20 |
| 15 | −0.98 | 1.38 | 1.11 | 0.27 | 0.31 | 0.31 | 0.29 |
| 16 | −1.11 | 0.20 | −0.37 | 0.18 | 0.21 | 0.40 | 0.23 |
| 17 | −3.08 | −2.25 | 0.27 | 0.09 | 0.14 | 0.13 | 0.11 |
| 18 | 3.56 | −0.63 | −0.26 | 0.55 | 0.32 | 0.52 | 0.49 |
| 19 | 8.45 | −1.31 | 2.31 | 0.85 | 0.48 | 0.77 | 0.76 |
| 20 | −1.46 | −0.08 | −1.51 | 0.17 | 0.17 | 0.39 | 0.21 |
| 21 | −3.06 | −0.36 | −1.03 | 0.09 | 0.26 | 0.13 | 0.13 |
| 22 | −1.37 | 1.62 | 0.44 | 0.16 | 0.35 | 0.33 | 0.24 |
| 23 | −1.48 | 0.01 | 2.22 | 0.14 | 0.30 | 0.42 | 0.23 |
| 24 | −0.46 | 3.03 | 1.82 | 0.25 | 0.45 | 0.36 | 0.31 |
| 25 | 2.09 | −1.11 | −0.65 | 0.37 | 0.26 | 0.47 | 0.36 |
| 26 | 2.13 | 0.73 | −2.83 | 0.38 | 0.26 | 0.54 | 0.38 |
| 27 | 1.82 | −1.43 | −0.48 | 0.33 | 0.40 | 0.44 | 0.36 |
| 28 | −0.60 | −0.98 | −2.15 | 0.14 | 0.19 | 0.50 | 0.22 |
| 29 | −0.42 | 0.05 | −2.14 | 0.17 | 0.28 | 0.42 | 0.24 |
| 30 | 7.07 | 0.44 | −3.09 | 0.74 | 0.32 | 0.79 | 0.66 |
| 31 | 1.81 | 3.11 | −0.07 | 0.43 | 0.47 | 0.45 | 0.44 |
| 32 | −0.42 | 1.48 | −1.96 | 0.26 | 0.32 | 0.31 | 0.28 |
| 33 | 3.83 | 2.51 | −0.41 | 0.63 | 0.37 | 0.45 | 0.54 |
| 34 | 2.99 | −3.51 | 2.80 | 0.45 | 0.15 | 0.59 | 0.41 |
| 35 | −0.46 | −3.01 | 0.14 | 0.34 | 0.06 | 0.23 | 0.26 |

CP, crude protein; CF, crude fiber; Glc, glucose; Fru, fructose; MA, malic acid; OA, oxalic acid; CA, citric acid; VC, vitamin C; TAA, total amino acid; CI, comprehensive indexes.

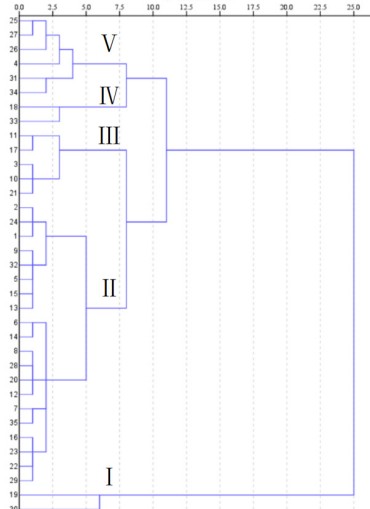

**Figure 3.** The dendrogram of clusters of 35 Chinese cabbage materials.

**Table 4.** Comparison of mineral element (mg/kg) and other nutrients (g/kg) of different groups of Chinese cabbage.

| Index | I | II | III | IV | V |
|---|---|---|---|---|---|
| CP | 15.90 | 15.75 | 12.93 | 13.17 | 11.10 |
| CF | 7.50 | 6.20 | 6.12 | 4.93 | 4.56 |
| Glc | 18.48 | 15.42 | 15.87 | 13.93 | 13.30 |
| Fru | 22.84 | 17.73 | 13.77 | 12.55 | 10.87 |
| MA | 1.73 | 0.82 | 0.96 | 0.76 | 0.68 |
| CA | 0.93 | 0.38 | 0.26 | 0.16 | 0.11 |
| OA | 0.16 | 0.10 | 0.12 | 0.09 | 0.07 |
| VC | 0.44 | 0.34 | 0.32 | 0.22 | 0.20 |
| TAA | 2.70 | 2.71 | 2.14 | 3.58 | 2.53 |
| Ca | 1153.17 | 1091.79 | 847.72 | 585.77 | 531.49 |
| K | 2235.44 | 2095.42 | 2156.55 | 1945.40 | 1563.52 |
| Mg | 164.60 | 164.29 | 137.10 | 120.31 | 97.95 |
| P | 430.14 | 448.97 | 408.25 | 391.71 | 356.44 |
| Cu | 0.86 | 0.57 | 0.69 | 0.42 | 0.40 |
| Fe | 16.74 | 13.26 | 9.55 | 5.87 | 4.75 |
| Mn | 2.39 | 1.69 | 1.51 | 1.24 | 1.08 |
| Zn | 2.15 | 2.31 | 2.17 | 2.26 | 1.57 |

CP, crude protein; CF, crude fiber; Glc, glucose; Fru, fructose; MA, malic acid; OA, oxalic acid; CA, citric acid; VC, vitamin C; TAA, total amino acid.

### 3.2.3. Screening of Evaluation Indexes for Nutritional Composition of Chinese Cabbage

In order to effectively analyze the corresponding relationship between 17 single indicators and the nutritional quality of different Chinese cabbage varieties, a multiple stepwise regression analysis method was used to take the comprehensive score of nutritional composition quality (D value) as the dependent variable and individual nutrition indicators as independent variables to establish the optimal mathematical model of the regression equation to predict nutritional quality. The Chinese cabbage nutritional quality identification of regression equation is as follows:

$$Y = 0.652 + 5.126Fe + 0.024CP + 0.419VC + 0.004Fru + 0.041MA + 0.307P + 51.036Mn + 0.969Mg + 62.092Cu + 0.028K.$$

This equation determines the coefficient $R^2 = 0.998$, $F = 1001.48$, reaching the level of extreme significance ($p < 0.001$), and the Durbin–Watson statistic $d = 1.744$; it fully shows that the model has strong explanatory ability. According to the formula, 10 of the 17 individual indexes were significantly correlated with the comprehensive score (D) of nutritional composition ($p < 0.05$), namely Fe, crude protein, vitamin C, fructose, malic acid, P, Mn, Cu and K. These 10 indexes can be used to evaluate the quality of the nutrient composition.

## 4. Discussion

Research on vegetable quality is shifting to multi-angle studies on flavor and nutrient content. As one of the most commonly consumed vegetables, Chinese cabbage will require increasingly higher nutritional quality. There is a strong correlation between the nutritional quality of Chinese cabbage and nutritional quality indexes such as vitamin C, soluble protein, soluble sugar and cellulose mass fraction [8]. Therefore, the content of each nutrient component is an important index to measure the quality of Chinese cabbage. According to the current research on vegetable nutritional quality at home and abroad, many researchers have screened out a variety of quality-related morphological, physiological, biochemical and metabolic nutrient composition indexes. In this study, 17 quality components of 35 Chinese cabbage materials were determined. Based on the 17 nutritional components, the quality characteristics of Chinese cabbage were evaluated using the membership function method and principal component analysis.

Vegetables are not the main way for people to get protein, but now more and more people are starting to follow a balanced diet, a healthy diet, reducing the intake of animal protein and paying more attention to vegetarian food, so people expect to receive good quality vegetable protein. Protein-rich foods, such as legumes, nuts, and seeds, can adequately provide the necessary proteins for adults following vegetarian or vegan diets [42]. The crude protein content of 35 Chinese cabbage materials in this experiment was 1.06~1.69%. In terms of the tested materials, the crude protein content of all of the Chinese cabbage varieties was above 1% (FW), and the protein contents of varieties 19, 24, 30, 31 and 33 were greater than 1.5%. Except for 19, all of the varieties were Qingmaye types, which was consistent with the results of previous studies, the protein content of Tianjin Qingmaye type is the highest [7] and provides a reference and ideas for the application of Chinese cabbage in food and improvement in the functional properties of protein, and it provides more options for vegetarians to get protein in their diet. Crude fiber has a negative effect on the sensory quality of Chinese cabbage [43]; however, due to the special physiological health function of dietary fiber [44,45], the quality breeding of cabbage should be improved, and if the taste quality is not significantly affected, the dietary fiber content should be increased as much as possible so that the total dietary fiber intake is maintained at an appropriate level. In this study, the average crude fiber content of the 35 Chinese cabbage samples was 0.53%, and the coefficient of variation among different materials was 17.86%, indicating that the crude fiber content of different materials varied greatly, and has great potential for improvement.

Sugar is an important organic compound in fruit and vegetables, consisting mainly of glucose, fructose and sucrose, and is an important indicator of the nutritional and sensory quality of vegetables [3]. A large number of studies have shown that glucose is an indispensable nutrient in the metabolism of many organisms, as well as being one of the main sources of calories. Fructose has a low glycemic index (GI) and is easily absorbed and utilized by the body. It is suitable for people with glucose metabolism and liver dysfunction and can be used to supplement energy [46]. Previous studies have shown that the soluble sugar content had the greatest influence on the quality of raw and cooked Chinese cabbage food through multiple stepwise regression analysis, and soluble sugar and soluble protein had a positive effect on sensory quality through path analysis, that is, the increase in soluble sugar and soluble protein content was conducive to the formation of good quality [3,43]. Meanwhile, in a previous study, using multiple regression analysis and path analysis, it was pointed out that the contribution value of glucose to the comprehensive flavor quality of Chinese cabbage is negative, but the contribution value of fructose is positive [4]. This result is consistent with the finding that the last 10 key indicators in this study include fructose. Therefore, the determination of the sugar component content is of great significance for nutritional quality evaluation. Two sugar components, glucose and fructose, were determined in 35 Chinese cabbage samples. The glucose content ranged from 10.09 to 21.14 g/kg, with an average of 14.52 g/kg. The content of fructose ranged from 7.72 to 28.26 g/kg, with an average value of 13.4 g/kg. In addition, the coefficients of variation of glucose and fructose among the different materials were 18.83% and 32.08%, respectively, indicating that the variations in glucose and fructose contents among the different materials was large, which has great potential for improvement.

The content of organic acids in vegetables is very small, but it plays an important role in the comprehensive evaluation of the nutritional quality of Chinese cabbage. The content and types of organic acids have an important impact on the nutritional quality and sensory quality of fruits and vegetables [47]. There are many types of organic acids; malic acid and citric acid are the most common, and different varieties of fruits and vegetables have their corresponding organic acids and contents [48,49]. Organic acids in fruits and vegetables have antibacterial, antiviral, blood sugar lowering, appetite-stimulating, digestive, and endocrine functions, which can enhance the absorption of other nutrients, promote metabolism, prevent diseases and fight cancer [50]. Previous studies have shown that organic acids have a negative effect on the cooked food quality of Chinese cabbage and also

contribute negatively to the overall flavor quality of Chinese cabbage [3,4]. Previous studies also found that an appropriate reduction in the organic acid content was an important trait for high-quality breeding [40]. In this study, three organic acids, malic acid, citric acid and oxalic acid, were detected in 35 varieties of Chinese cabbage, and the average content of organic acids in 35 varieties of Chinese cabbage was about 1.16 g/kg. The correlation analysis results of organic acids showed that the correlation among malic acid, citric acid and oxalic acid reached a significant level; in addition, their coefficients of variation among the different materials were 54.34%, 91.47% and 41.97%, respectively, indicating that the content of organic acids varied greatly among different materials, which would be of great significance for the improvement and breeding of special materials. Among them, malic acid is one of the 10 key positive indicators to evaluate the nutritional quality of Chinese cabbage, which is inconsistent with the results of previous studies on organic acids in the flavor and sensory quality of Chinese cabbage. In addition to the independent variable (nutrient composition), the difference or the range and quantity of materials selected, is it also related to human subjectivity? As the evaluation of flavor and sensory quality requires subjective evaluation, it needs to be further discussed.

Vitamin C is a fundamental nutrient essential for the human body [51]. It synthesizes connective tissue, particularly collagen, and plays a crucial role as an antioxidant by scavenging and neutralizing oxidants within the body [52,53]. The content of VC in Chinese cabbage is one of the important indicators of its nutritional value. Previous studies have found that VC has no significant effect on the sensory quality and overall flavor quality of Chinese cabbage, but appropriately increasing the VC content is of great importance for high-quality breeding. In this study, the average content of VC was about 255 mg/kg; compared with the VC content (47 mg/100 g in Chinese cabbage) reported in the China Food Composition Table 2015 Complete Edition, the average VC content of Chinese cabbage in this study was lower, but the coefficient of variation between different Chinese cabbage materials was 31.26%. This shows that the variation in vitamin C content between different materials is large and has great potential for improvement.

Mineral elements required by the human body are obtained directly and indirectly from the external environment, which are cultivated from various crops., and the human body needs a certain amount of minerals every day to maintain physiological functions and biochemical metabolism [54]. Plants have different requirements for different elements, and the element content is affected by the variety of genotypes and environmental factors [55]. An analysis of the mineral content in 91 bottle gourd varieties unveiled considerable diversity and noteworthy genotypic distinctions in these samples [56]. The analysis of the mineral content of 61 Camellia sinensis germplasm resources in China showed that 18 kinds of mineral elements had a high coefficient of variation in these materials, indicating that this Camellia sinensis germplasm had rich variation types in mineral elements [57].Therefore, the contents of different elements in the same cabbage material are different, and the contents of the same element in different cabbage materials are also different. The content of K, Ca, P and Mg exceeded 100 mg/kg (FW), and the average content of K was 1952.19 mg/kg. The content of Fe, Zn, Mn and Cu was less than 10 mg/kg, and the content of Cu was less than 1 mg/kg. The content of mineral elements in Chinese cabbage showed a large range of variation, with the coefficients of variation between 11.45% and 49.83%; the coefficient of variation for Ca, Cu and Fe was more than 30%, which was a large range of variation in the Chinese cabbage and could be screened out from the ideal material. In this study, K, P, Mg, Cu, Fe and Mn were the key indicators for screening the quality of Chinese cabbage, and this result was partially consistent with the previous study on walnuts, which found that Mg was the key factors for comparing the quality differences among different durian [58]. Other studies have found that Ca, Mg, Na, Ni, Cu, As, Mn, P, Zn, K and Fe are important traits for evaluating the mineral element content of rice [59].

The types and contents of amino acids directly affect the flavor quality of vegetables and are also important indicators for evaluating their nutritional quality [60,61]. Amino acids have crucial functions as primary metabolites and fundamental components of plant

proteins, playing an indispensable role in plant growth, development, and reproduction [62]. They are the foundation for additional biosynthetic pathways and are crucial in signal transduction, plant response to stress, and protein synthesis [63]. The human body requires nine essential amino acids, namely lysine (Lys), tryptophan (Trp), phenylalanine (Phe), methionine (Met), threonine (Thr), isoleucine (Ile), leucine (Leu), and valine (Val), which are acquired from food. Previous studies have shown that there is a significant positive correlation between full flavor and amino acid content, suggesting that the level of this trait is closely related to umami flavor [40]. In this study, 22 kinds of free amino acids were detected in Chinese cabbage, including all 8 kinds of essential amino acids, indicating that amino acids in Chinese cabbage are rich and diverse. In conclusion, although total free amino acid is not the key index to evaluate the nutritional quality of Chinese cabbage in this study, it is still important for the nutritional quality of Chinese cabbage.

The results of the correlation analysis between different quality characters can be used as an important reference for the selection of and improvement in a variety of quality characters. Willems et al. [64] analyzed the correlation between Ca, Zn, Fe, K, Mg and other elements in the F2 generation of Arabidopsis hybrids, and showed that Ca and Mg had extremely significant positive correlations, and Zn had extremely significant positive correlation with Fe and Ca. In the comprehensive evaluation of crop nutritional quality, the correlation between quality traits can be used to select relatively simple and easy-to-measure traits as indicators, in order to improve the screening efficiency of early breeding. In this study, we also found that there were significant or extremely significant ($p < 0.05$ or $p < 0.01$) correlations between different nutritional elements of Chinese cabbage in many groups, either significant positive correlations or significant negative correlations. These results indicated that the correlations between different elements were affected by the variety, and the correlations between these quality components played an important role in studying the genetic rule of nutrients in Chinese cabbage and simplified the process of nutrient quality analysis. Membership function and principal component analyses were used to rank the nutritional quality of different varieties. Through cluster analysis, 35 D values of Chinese cabbage variety quality were clustered and divided into five grades. Among the 35 varieties, the varieties with high nutritional quality were divided into two categories (D ≥ 0.66, 0.49 ≤ D ≤ 0.54), and there were four varieties in total, which reflected the difference between them and other categories of Chinese cabbage; the quality components in this category were more abundant. Finally, according to D values, the mathematical model of the 10 key evaluation indexes of the nutritional quality of Chinese cabbage was established by a stepwise regression method, which was beneficial to quickly identify the nutritional quality of Chinese cabbage.

## 5. Conclusions

In order to make the results more intuitive and reliable, multivariate statistical techniques including diversity analysis, correlation analysis, cluster analysis and principal component analysis, were used to process the data of different nutrient compositions. The coefficient of variation of 17 kinds of nutrients was different, and the coefficient of variation of CA was the largest, reaching 91.47%. This indicated that CA was most susceptible to Chinese cabbage varieties. The correlation analysis of 17 nutrients showed that there were several groups with significant or extremely significant positive correlations among them, and the correlation coefficients were all very high. Membership function and principal component analyses were used to rank the nutritional quality of different varieties. A cluster analysis was carried out on the 35 D values of Chinese cabbage varieties for quality clustering, divided into five grades. Group I had the best nutritional quality, and group V had the worst nutritional quality, correspondingly. Meanwhile, four varieties in groups I and II had higher levels of each nutrient quality than varieties in the other three groups. A stepwise regression method was used to establish the Chinese cabbage's nutritional quality, which included a mathematical model of 10 key evaluation indexes, and this helped to quickly identify the nutritional qualities of Chinese cabbage. This study can

provide a theoretical basis for consumers to choose Chinese cabbage varieties with high nutritional quality, and also provide guidance for improving the nutritional quality of Chinese cabbage varieties.

**Author Contributions:** Writing—original draft, conceptualization, formal analysis, C.S.; data curation, X.Y.; investigation and methodology, G.L. (Guangyang Liu); Supervision, S.Z. (Shifan Zhang), G.L. (Guoilang Li) and R.S.; writing—review and editing, F.L., H.Z. and C.W.; Project administration and Funding acquisition, D.X. and S.Z. (Shujiang Zhang). All authors have read and agreed to the published version of the manuscript.

**Funding:** This research was funded by the China Agriculture Research System (CARS-23-A-14), Central Public-interest Scientific Institution Basal Research Fund (No. Y2023XK02) and the Agricultural Science and Technology Innovation Program of the Chinese Academy of Agricultural Sciences (CAAS-ASTIP-IVFCAAS). This research was performed at the State Key Laboratory of Vegetable Biobreeding, Institute of Vegetables and Flowers, the Chinese Academy of Agricultural Sciences, Beijing 100081, China, and the Key Laboratory of Biology and Genetic Improvement of Horticultural Crops, the Ministry of Agriculture, Beijing, China.

**Data Availability Statement:** Data are contained within the article or Appendix A.

**Conflicts of Interest:** The authors declare that they have no known competing financial interests or personal relationships that could have appeared to influence the research reported in this study.

## Appendix A

**Table A1.** The germplasm resources of 35 Chinese cabbage materials.

| Number | Materials Name | Pedigrees | Number | Materials Name | Pedigrees |
|---|---|---|---|---|---|
| 1 | 1911462 | TianFu75 (NongBoDa) | 19 | 1911679 | BaoHongXin 24 |
| 2 | 1911606 | HongHaiEr (XianZhengDa) | 20 | 1911504 | heatwave 539f1-2 |
| 3 | ZB61 | F1 | 21 | 1911556 | JuLongKangReXianFeng |
| 4 | 1616030 | XY4A | 22 | LS70 | F1 |
| 5 | 1911507 | ZhenBao50 | 23 | 1911847 | TaQing07 |
| 6 | JHWW | F1 | 24 | 1911421 | XiaoBaoJian |
| 7 | 1911095 | 127WaWaCai (XY) | 25 | 1911754 | YuTianBaoJian × TaQing |
| 8 | 1911013 | MiNiHuang (South Korea longjing) | 26 | 1911757 | YuTianBaoJian × TaQing |
| 9 | 1911014 | MiNiHuang (South Korea longjing) | 27 | 1915053 | CMSJinQiu × XiaoBaoJian |
| 10 | 1915154 | CR–LiMin | 28 | 1915054 | CMSJinQiu × XiaoBaoJian |
| 11 | 1915157 | CR–ZhongLianJinBao | 29 | 1915320 | ShunYi30 |
| 12 | 1915169 | GaoShanWaWaCai | 30 | 1911792 | 234LAangFang |
| 13 | 1911156 | JinJiang45 | 31 | 1911801 | BP058 |
| 14 | 1840414 | F1 | 32 | ZB76 | F1 (BP058 × 234) |
| 15 | A04749 | F1 | 33 | 1911830 | JinLv75 |
| 16 | 1640250 | F1 | 34 | 1914040 | DongLiKuaiCai |
| 17 | JH308 | F1 | 35 | 1914078 | FuHuaKuaiCai |
| 18 | 1911676 | BaoHongXin24 | | | |

**Table A2.** Methods for the determination of different components.

| Components | Determination Methods |
|---|---|
| vitamin C | 2,6-dichloroindophenol titration (GB 5009.86-2016) |
| crude protein | Combustion Nitrogen Analysis (GB 5009.5-2016) |
| crude fiber | Acid—alkali washing method (GB 5009.10-2003) |
| glucose, fructose | Ion chromatography |
| malic acid, citric acid, oxalic acid | Ion chromatography (NYT2277-2012) |
| mineral elements | ICP-OES (GB 5009.268) |
| amino acid | amino acid analyzer (GB/T 30987-2020) |

1.   Vitamin C:

•   Test solution preparation: Weigh 100 g of the edible portion of the sample and place it in a pulverizer. Add 100 g of metaphosphoric acid solution and quickly pound it until homogenous. Accurately weigh 10–40 g of homogenized sample (to an accuracy of

0.01 g) into a beaker, transfer to a 100 mL volumetric flask, and dilute to the mark with a metaphosphoric acid solution. Shake well and filter.

- Titration: Transfer 10 mL of the filtrate to a 50 mL conical flask with precision, and titrate it with calibrated 2,6-dichloroindophenol solution until the solution remains pink for 15 s without any fading. It is critical to perform a blank test simultaneously.
- Calculation of results:

$$x = \frac{(V - V_0) \times T \times A}{m} \times 100$$

*x*—the content of L(+) ascorbic acid in the sample, in mg/100 g;
*V*—volume of 2,6 dichloroindophenol solution consumed in the titration of the sample, in milliliters (mL);
$V_0$—volume of 2,6 dichloroindophenol solution consumed in the titration blank, in milliliters (mL).
*T*—titration of 2,6 dichloroindophenol solution, expressed as milligrams of ascorbic acid per milliliter of 2,6 dihloroindophenol solution (mg/mL).
*A*—dilution factor;
*m*—mass of the sample, in grams (g).

2.  Crude protein

- Weigh 0.1 g~1.0 g fully mixed sample (accurate to 0.0001 g) according to the instructions of the instrument, wrap it with tin foil and place it on the sample tray. After entering the combustion reactor (900 °C~1200 °C), the sample is fully burned in high-purity oxygen (≥99.99%). The product (NOx) in the combustion furnace is transported by the carrier gas carbon dioxide or helium to the reduction furnace (800 °C), and its content is measured after reduction to produce nitrogen.
- Calculation of results:

$$x = C \times F$$

*x*—the protein content in the sample, expressed in grams per hundred grams (g/100 g);
*C*—the content of nitrogen in the sample, expressed in grams per hundred grams (g/100 g);
*F*—coefficient of conversion of nitrogen to protein.

3.  Crude fiber

- Weigh 20 g~30 g of mashed specimen, transfer to 500 L-shaped flask, add 200 mL of boiling 1.25% sulphuric acid, heat to bring to a slight boil, keep the volume constant, maintained for 30 min, and shake the conical flask every 5 min, in order to fully mix the substances in the flask.
- The conical flask was removed and immediately filtered using linen cloth and washed with boiling water until the washings were not acidic.
- Then, use 200 mL of boiling 1.25% potassium hydroxide solution to wash the residue on the linen into the original conical flask and heat for 30 min, remove the conical flask, immediately filter using linen, wash with boiling water for 2~3 times, transfer to a dry and weighed G2 pendant crucible or pendant funnel of the same type, extract the filter, wash with hot water, and then pump dry. The crucible is washed once more with ethanol and ether. Dry the crucible and contents in an oven at 105 °C and weigh, repeating the operation until a constant amount is obtained.
- If the sample contains more insoluble impurities, the sample can be moved to the asbestos crucible, dried and weighed, and then moved to the 550 °C high-temperature furnace ashing; all the carbon-containing material ashing was placed in the desiccator, cooled to room temperature and weighed, and the amount of loss of crude fiber content was calculated.
- Calculation of results:

$$x = \frac{G}{m} \times 100\%$$

*x*—the content of coarse fibers in the sample;
*G*—the mass of the residue (or mass lost by a high-temperature furnace), expressed in grams (g);
*m*—the mass of the sample in grams (g).

4. Glucose and fructose

- Homogenate 5.00 g was extracted using 80% ethanol at a constant volume of 50 mL for 30 min by ultrasound or shaker oscillator. After centrifugation at 3000 r/min for 10 min, take 1 mL of the supernatant and put it into a 100 mL volumeter bottle, dilute it with water to the scale, and shake well. Finally, the diluent was directly injected into 0.22 μm aqueous filtration membrane and C18 solid phase extraction column for analysis.
- Column parameters: anion exchange sugar protection column CarboPac PA 10 (50 mm × 4 mm); anion exchange sugar analysis column CarboPac PA 10 (250 mm × 4 mm). NaOH gradient leaching, flow rate: 0.80 mL/min. Sample size: 10 μL. Column temperature: 30 °C. Amperometric detector: Gold working electrode, Ag/AgCl reference electrode mode, and sugar standard four-potential waveform pulse amperometric detection.
- The sample treatment solution and the standard working solution were separately injected into the ion chromatograph for separation and detection. The sample solution was qualitatively determined using the retention time of the standard solution peak and quantitatively determined using the area of the standard solution peak.
- Calculation of results:

$$x = \frac{(C - C_0) \times V \times f}{m \times 1000 \times 10}$$

*x*—the content of fructose and glucose in the sample, expressed in grams per hundred grams or grams per hundred milliliters (g/100 g or g/100 mL);
*C*—the concentration of fructose and glucose in the sample solution calculated from the standard curve, in mg/L;
$C_0$—the concentration of fructose and glucose in the blank calculated from the standard curve, in milligrams per liter (mg/L);
*V*—the volume of the constant volume, in milliliters (mL);
*m*—the weight of the sample, expressed in grams (g) or milliliters (mL);
*f*—dilution ratio;
10—conversion factor;
1000—conversion factor.

When the sample size was 2 g, the LOD of fructose and glucose was 0.015 g/100 g, and the LOQ was 0.050 g/100 g.

1. Malic acid, citric acid and oxalic acid

- Sample preparation: The edible part is extracted according to the provisions of GB/T 8855; after it is reduced, it is chopped, thoroughly mixed and crushed in the food processor to produce the test sample.
- Withdraw: Weigh 5 g (accurate to 0.001 g) sample in a 100 mL beaker, add 80 mL water, put it into an ultrasonic meter, followed by ultrasonic treatment for 30 min, transfer it to a 100 m volumetric bottle and maintain constant volume with water, fully mixed; 0.22 μm of the water phase filter membrane was ready to be measured.
- Instrument: Column: High volume anion exchange column, such as AS19, or other columns with comparable performance; column temperature: 30 °C; the sample size

was 25 µL; mobile phase: potassium hydroxide solution was used as eluent at a flow rate of 1.0 mL/min.

- Standard curve drawing: The standard curve was drawn using the mass concentration of the standard series solution as the horizontal coordinate and the peak area as the vertical coordinate.
- Test solution determination: The retention time was used for qualitative analysis, and the peak area of the test solution and the standard working solution was compared for quantitative analysis.
- Result calculation:

$$x = \frac{\rho \times V \times 1000}{m \times 1000}$$

$x$—the content of Malic acid, citric acid and oxalic acid in the sample, expressed in in milligrams per kilogram (mg/kg);
$\rho$—The mass concentration of components to be measured in the test solution was obtained by linear regression equation, and the unit was mg/L.
$V$—Constant volume unit, in milliliters (mL).
$m$—Sample mass, in grams (g).

The LOD of this standard method was 2.0 mg/kg~9.0 g/kg.

6. Mineral elements

- Sample preparation: homogenize the edible part of the sample.
- Sample digestion: Accurately weigh 0.5 g~5 g (accurate to 0.001 g) or accurately remove 2.00 mL~10.0 mL of the sample into glass or Teflon containers. In the solution vessel, the samples containing ethanol or carbon dioxide are first heated on the electric heating plate at low temperature to remove ethanol or carbon dioxide, and 10 mL of nitric acid–perchloric acid (10 + 1) mixed solution is added and then digested on the electric heating plate or graphite digestion device. If the digestion solution turns brown and black during digestion, a small amount of mixed acid can be added appropriately until white smoke is emitted, and the digestion solution is colorless, transparent or slightly yellow and cold. Next, add 25 mL or 50 mL of water, mix well and set aside; perform a blank test at the same time.
- Instrument operation reference conditions: Observation method: vertical observation— if the instrument has a two-way observation method, high-concentration elements, such as potassium, sodium, calcium, magnesium and other elements, should be observed vertically, and the rest should be observed horizontally.
- Power: 1150 W; plasma gas flow rate: 15 L/min; auxiliary gas flow rate: 0.5 L/min; atomizing gas flow rate: 0.65 L/min; analysis pump speed: 50 r/min.
- Production of standard curves: The standard series of working solutions was injected into the inductively coupled plasma emission spectrometer, and the intensity signal response value of the analytical spectral line of the element to be measured was determined. The concentration of the element to be measured was the horizontal coordinate, the intensity response value of the analytical spectral line was the longitudinal coordinate, and the standard curve was drawn.
- Determination of sample solution: The blank solution and the sample solution were injected into the inductively coupled plasma emission spectrometer to measure the signal response of the analysis spectral line intensity of the element to be measured, and the concentration of the element to be measured in the digestion solution was obtained according to the standard curve.
- Result calculation:

$$x = \frac{(\rho - \rho_0) \times V \times f}{m}$$

*x*—the content of elements to be measured in the sample, expressed in milligrams per kilogram or milligrams per liter (mg/kg or mg/L).

$\rho$—mass concentration of the element to be measured in the sample solution, in mg/L.

$\rho_0$—Mass concentration of the element to be measured in the blank solution of the sample, in milligrams per liter (mg/L).

*V*—constant volume of the digestive fluid of the sample, in milliliters (mL).

*f*—dilution ratio of the sample.

*m*—sample weighed by mass or removed volume in grams or milliliters (g or mL).

**Table A3.** LOD and LOQ of inductively coupled plasma emission spectrometry (ICP-OES) and the recommended analytical spectral lines of the elements to be measured.

| Element Symbol | The Wave Length of Analytical Line (nm) | LOD1 (mg/kg) | LOD2 (mg/L) | LOQ1 (mg/kg) | LOQ2 (mg/L) |
|---|---|---|---|---|---|
| Ca | 315.8/317. | 5 | 2 | 20 | 5 |
| Cu | 324.75 | 0.2 | 0.05 | 0.5 | 0.2 |
| Fe | 239.5/259.9 | 1 | 0.3 | 3 | 1 |
| K | 766.49 | 7 | 3 | 30 | 7 |
| Mg | 279.079 | 5 | 2 | 20 | 5 |
| Mn | 257.6/259. | 0.1 | 0.03 | 0.3 | 0.1 |
| P | 213.6 | 1 | 0.3 | 3 | 1 |
| Zn | 206.2/213. | 0.5 | 0.2 | 2 | 0.5 |

7. Amino acid

- The samples were prepared according to 6.1 and 6.2 in GB/T8303-2013 and passed through a 40-mesh sieve. Mix well and put into a clean container as a sample.
- Test procedure:
- Standard solution composition: Preparation of mixed amino acid standard reserve solution. Weigh an appropriate amount of each amino acid standard (slightly to 0.01 mg) and dissolve in water to prepare a mixed solution. The concentrations of theanine and other amino acids in the mixed standard solution were 5.00 μmol/mL and 1.25 μmol/mL, respectively. The shelf life of the frozen solution is 1 month.
- Mixed amino acid standard working liquid preparation: Accurately absorb 2 mL of the standard reserve liquid of mixed amino acids into a 5 mL volumetric bottle, dilute the volume with water to the scale, and mix well to obtain the first standard solution. The first standard solution is diluted step by step with water to produce a total of 7 different concentrations of the series of mixed standard working solutions. The concentrations of theanine and other basic acids in the mixed standard working solution of 7 concentrations were 2000.00 nmol/mL and 500.00 nmol/mL, 1000.00 nmol/mL and 250.00 nmol/mL, respectively. A total of 500.00 nmol/mL and 125.00 nmol/mL, 250.00 nmol/mL and 62.50 nmol/mL,125.00 nmol/mL and 31.25 mol/mL, 62.50 mol/mL and 15.63 nmol/mL, 31.25 nmol/mL and 7.81 nmol/mL, were ready for use.
- The mobile phase and the post-column derivatization reaction solution were prepared.
- Configure the mobile phase B1, B2, B3, B4, B5 and B6, and configure the reaction solution R1, R2 and R3.
- Sample extraction: Accurately weigh the sample, which is about 20 g (accurate to 0.0001 g), add 200 mL boiling water into a 250 mL cone, heat it in a 95 °C water bath, mix it well every 5 min, extract it for 10 min, and then filter it while it is hot. After the filtrate is cooled to room temperature, fill it with water to 250 mL, mix it well and take an appropriate amount of sample solution. After filtration through a 0.45 μm water phase filter membrane, it is ready to be measured.
- Measurement: Column: sulfonic acid type cation exchange column, 3 μm, 4.6 mm × 60 mm, or equivalent performance column. Instrument separation system operation reference conditions: The flow rate of the mobile phase was 0.35 mL/min, and the sample vol-

ume was 20 μL. Amino acid reaction detection system operation reference conditions were as follows:

- The temperature of the reaction column was 135 °C, the flow rate of the reaction liquid was 0.30 mL/min, and the detection wavelengths of the detector were 570 nm and 440 nm.
- Draw a standard working curve: The automatic amino acid analyzer was started and the working parameters were set. After the baseline was stabilized, a series of mixed amino acid standard working solutions of different concentrations were absorbed and injected into the automatic amino acid analyzer for determination, and the peak areas of different amino acids were obtained, respectively. The standard working curve was established using the peak area of each amino acid as the vertical coordinate and the concentration as the horizontal coordinate. The peak order of each amino acid was identified by retention time.
- Sample determination: The chromatographic peak area of each amino acid in the sample was obtained using the amino acid analyzer, and the content was calculated from the standard working curve. Using water as a blank sample, the amino acid background values in the blank sample were calculated under the same conditions. The net amino acid content in each sample was obtained by deducting the amino acid background value in the blank sample.
- Result calculation:

$$W_1 = \frac{(C - C_k) \times V \times M \times 10^{-6}}{m \times m_1} \times 100$$

$W_1$—The content of each amino acid component in a sample, expressed in milligrams per 100 g (mg/100 g);
$C$—The amino acid concentration calculated from the standard working curve in the sample solution, in units of nanomolar per milliliter (nmol/mL);
$C_k$—The amino acid concentration calculated from the standard working curve in the blank sample solution, expressed in nanomolar per milliliter (nmol/mL);
$V$—Total sample volume, in milliliters (mL);
$M$—The molar mass of an amino acid, expressed in grams per mole (g/mol);
$m$—Sample mass, expressed in grams (g);
$m_1$—The dry matter rate of the sample was determined by GB/T8303.

The LOQ of this standard method is 13.44 μg/kg~296.31 μg/kg.

**Table A4.** Results of 15 quality components of 35 Chinese cabbage (except Mineral element and TAA).

| Materials Number | DW % | VC mg/kg | CP % | CF % | Glc g/kg | Flu g/kg | MA g/kg | CA g/kg | OA g/kg |
|---|---|---|---|---|---|---|---|---|---|
| 1 | 5.53 ± 0.00 | 230.37 ± 3.19 | 1.35 ± 0.02 | 0.55 ± 0.03 | 11.05 ± 0.08 | 8.53 ± 0.03 | 0.93 ± 0.04 | 0.09 ± 0.01 | 0.18 ± 0.03 |
| 2 | 5.73 ± 0.00 | 281.32 ± 37.36 | 1.36 ± 0.02 | 0.52 ± 0.00 | 11.61 ± 0.66 | 9.20 ± 0.87 | 0.90 ± 0.13 | 0.11 ± 0.01 | 0.26 ± 0.05 |
| 3 | 4.86 ± 0.00 | 186.49 ± 16.09 | 1.10 ± 0.04 | 0.45 ± 0.01 | 11.31 ± 0.62 | 8.54 ± 0.51 | 1.04 ± 0.12 | 0.14 ± 0.02 | 0.19 ± 0.03 |
| 4 | 6.45 ± 0.00 | 351.77 ± 32.32 | 1.21 ± 0.03 | 0.59 ± 0.01 | 14.83 ± 0.89 | 9.44 ± 1.15 | 0.75 ± 0.49 | 0.14 ± 0.02 | 0.29 ± 0.02 |
| 5 | 5.24 ± 0.00 | 182.35 ± 11.20 | 1.42 ± 0.03 | 0.45 ± 0.03 | 11.20 ± 0.48 | 7.72 ± 0.50 | 0.82 ± 0.16 | 0.13 ± 0.02 | 0.13 ± 0.04 |
| 6 | 5.11 ± 0.00 | 206.20 ± 12.39 | 1.06 ± 0.05 | 0.46 ± 0.01 | 13.35 ± 0.85 | 15.30 ± 1.07 | 0.71 ± 0.04 | 0.08 ± 0.02 | 0.11 ± 0.02 |
| 7 | 5.09 ± 0.01 | 211.11 ± 34.29 | 1.48 ± 0.11 | 0.44 ± 0.02 | 11.30 ± 3.35 | 9.33 ± 2.87 | 0.54 ± 0.15 | 0.09 ± 0.08 | 0.12 ± 0.04 |
| 8 | 5.26 ± 0.00 | 158.16 ± 8.55 | 1.46 ± 0.01 | 0.47 ± 0.03 | 14.66 ± 1.25 | 9.92 ± 1.82 | 0.49 ± 0.07 | 0.06 ± 0.02 | 0.11 ± 0.01 |
| 9 | 5.21 ± 0.00 | 196.27 ± 35.13 | 1.48 ± 0.10 | 0.45 ± 0.04 | 14.68 ± 0.95 | 9.41 ± 1.15 | 0.75 ± 0.22 | 0.06 ± 0.02 | 0.13 ± 0.05 |
| 10 | 5.05 ± 0.00 | 228.45 ± 10.61 | 1.15 ± 0.00 | 0.49 ± 0.03 | 18.35 ± 1.59 | 12.12 ± 2.09 | 0.45 ± 0.02 | 0.04 ± 0.00 | 0.06 ± 0.01 |
| 11 | 4.34 ± 0.01 | 189.42 ± 9.32 | 1.13 ± 0.14 | 0.44 ± 0.01 | 13.68 ± 0.34 | 12.78 ± 0.98 | 0.76 ± 0.14 | 0.06 ± 0.01 | 0.11 ± 0.04 |
| 12 | 5.17 ± 0.00 | 204.03 ± 10.09 | 1.24 ± 0.21 | 0.44 ± 0.01 | 15.65 ± 4.77 | 13.88 ± 2.73 | 0.84 ± 0.20 | 0.11 ± 0.06 | 0.19 ± 0.06 |
| 13 | 4.90 ± 0.00 | 173.68 ± 10.61 | 1.45 ± 0.04 | 0.42 ± 0.03 | 12.14 ± 0.78 | 12.82 ± 0.87 | 0.95 ± 0.14 | 0.07 ± 0.01 | 0.26 ± 0.03 |
| 14 | 5.26 ± 0.00 | 219.55 ± 8.56 | 1.25 ± 0.01 | 0.49 ± 0.02 | 13.97 ± 4.10 | 10.84 ± 3.78 | 0.54 ± 0.25 | 0.06 ± 0.02 | 0.12 ± 0.04 |
| 15 | 5.14 ± 0.00 | 222.29 ± 7.54 | 1.38 ± 0.22 | 0.45 ± 0.03 | 12.98 ± 0.78 | 11.05 ± 1.18 | 0.89 ± 0.10 | 0.08 ± 0.01 | 0.20 ± 0.07 |
| 16 | 5.44 ± 0.00 | 204.26 ± 11.00 | 1.25 ± 0.03 | 0.43 ± 0.00 | 15.67 ± 0.77 | 17.52 ± 0.47 | 0.89 ± 0.10 | 0.09 ± 0.04 | 0.17 ± 0.04 |
| 17 | 4.67 ± 0.00 | 181.61 ± 11.62 | 1.06 ± 0.04 | 0.47 ± 0.02 | 10.09 ± 1.09 | 8.53 ± 0.86 | 0.75 ± 0.04 | 0.09 ± 0.01 | 0.11 ± 0.01 |
| 18 | 6.95 ± 0.00 | 331.07 ± 13.01 | 1.45 ± 0.12 | 0.64 ± 0.04 | 16.04 ± 3.81 | 18.23 ± 3.85 | 1.03 ± 0.89 | 0.13 ± 0.03 | 0.35 ± 0.20 |
| 19 | 7.52 ± 0.00 | 344.58 ± 35.18 | 1.66 ± 0.05 | 0.70 ± 0.07 | 15.83 ± 0.66 | 17.41 ± 0.35 | 2.36 ± 0.31 | 0.18 ± 0.02 | 0.68 ± 0.13 |
| 20 | 5.73 ± 0.01 | 243.97 ± 14.04 | 1.26 ± 0.05 | 0.52 ± 0.03 | 19.00 ± 3.61 | 14.72 ± 3.56 | 0.63 ± 0.23 | 0.05 ± 0.02 | 0.12 ± 0.07 |
| 21 | 4.60 ± 0.00 | 201.98 ± 18.11 | 1.10 ± 0.05 | 0.42 ± 0.01 | 13.06 ± 1.53 | 12.36 ± 1.65 | 0.41 ± 0.06 | 0.04 ± 0.01 | 0.07 ± 0.01 |
| 22 | 6.00 ± 0.00 | 182.83 ± 0.00 | 1.34 ± 0.06 | 0.50 ± 0.01 | 14.54 ± 1.10 | 13.08 ± 1.05 | 0.91 ± 0.11 | 0.10 ± 0.02 | 0.10 ± 0.02 |
| 23 | 5.52 ± 0.00 | 230.37 ± 13.18 | 1.24 ± 0.06 | 0.48 ± 0.03 | 12.60 ± 1.08 | 11.31 ± 0.89 | 1.42 ± 0.06 | 0.19 ± 0.01 | 0.12 ± 0.01 |
| 24 | 5.89 ± 0.00 | 252.07 ± 28.53 | 1.53 ± 0.09 | 0.55 ± 0.02 | 12.22 ± 1.31 | 10.01 ± 1.47 | 0.85 ± 0.05 | 0.13 ± 0.01 | 0.22 ± 0.01 |

**Table A4.** *Cont.*

| Materials Number | DW % | VC mg/kg | CP % | CF % | Glc g/kg | Flu g/kg | MA g/kg | CA g/kg | OA g/kg |
|---|---|---|---|---|---|---|---|---|---|
| 25 | 6.13 ± 0.00 | 321.90 ± 34.77 | 1.15 ± 0.01 | 0.62 ± 0.06 | 14.99 ± 1.00 | 15.06 ± 1.37 | 0.81 ± 0.59 | 0.11 ± 0.02 | 0.22 ± 0.05 |
| 26 | 7.19 ± 0.01 | 353.98 ± 74.43 | 1.32 ± 0.22 | 0.68 ± 0.06 | 21.14 ± 1.38 | 17.58 ± 1.12 | 0.22 ± 0.15 | 0.06 ± 0.03 | 0.14 ± 0.10 |
| 27 | 6.06 ± 0.00 | 288.68 ± 29.36 | 1.24 ± 0.07 | 0.65 ± 0.09 | 14.50 ± 2.04 | 14.80 ± 2.23 | 0.63 ± 0.44 | 0.12 ± 0.04 | 0.20 ± 0.06 |
| 28 | 6.29 ± 0.00 | 268.06 ± 9.09 | 1.10 ± 0.02 | 0.61 ± 0.08 | 17.94 ± 0.96 | 18.99 ± 1.52 | 0.52 ± 0.36 | 0.13 ± 0.04 | 0.14 ± 0.07 |
| 29 | 6.35 ± 0.00 | 245.15 ± 8.65 | 1.18 ± 0.04 | 0.64 ± 0.06 | 16.43 ± 0.46 | 17.45 ± 0.08 | 0.57 ± 0.04 | 0.08 ± 0.01 | 0.11 ± 0.01 |
| 30 | 9.03 ± 0.00 | 536.32 ± 94.77 | 1.52 ± 0.01 | 0.80 ± 0.05 | 21.12 ± 2.36 | 28.26 ± 2.68 | 1.10 ± 0.03 | 0.13 ± 0.04 | 1.18 ± 0.34 |
| 31 | 6.39 ± 0.00 | 214.53 ± 44.55 | 1.66 ± 0.06 | 0.62 ± 0.03 | 17.29 ± 2.72 | 13.69 ± 2.13 | 0.91 ± 0.37 | 0.06 ± 0.02 | 0.27 ± 0.10 |
| 32 | 5.72 ± 0.00 | 207.91 ± 17.19 | 1.28 ± 0.11 | 0.54 ± 0.01 | 15.07 ± 0.70 | 19.48 ± 0.90 | 0.26 ± 0.01 | 0.05 ± 0.02 | 0.22 ± 0.04 |
| 33 | 6.70 ± 0.00 | 343.25 ± 23.28 | 1.69 ± 0.17 | 0.60 ± 0.03 | 14.79 ± 1.51 | 17.23 ± 2.02 | 0.60 ± 0.50 | 0.07 ± 0.03 | 0.41 ± 0.28 |
| 34 | 5.97 ± 0.00 | 378.03 ± 6.87 | 1.17 ± 0.01 | 0.50 ± 0.01 | 12.44 ± 0.55 | 12.05 ± 0.64 | 2.43 ± 0.24 | 0.19 ± 0.03 | 0.43 ± 0.05 |
| 35 | 5.28 ± 0.00 | 352.83 ± 7.94 | 1.16 ± 0.02 | 0.44 ± 0.01 | 12.58 ± 0.69 | 10.35 ± 0.62 | 0.77 ± 0.06 | 0.09 ± 0.02 | 0.16 ± 0.06 |

**Table A5.** Results of mineral elements in 35 Chinese cabbage.

| Materials Number | Ca mg/kg | K mg/kg | Mg mg/kg | P mg/kg | Cu mg/kg | Fe mg/kg | Mn mg/kg | Zn mg/kg |
|---|---|---|---|---|---|---|---|---|
| 1 | 677.71 ± 32.37 | 2166.25 ± 103.88 | 146.00 ± 6.15 | 422.29 ± 27.92 | 0.53 ± 0.00 | 6.67 ± 1.76 | 1.39 ± 0.08 | 2.70 ± 0.11 |
| 2 | 666.83 ± 111.07 | 2025.56 ± 155.54 | 127.77 ± 11.95 | 412.41 ± 10.02 | 0.50 ± 0.06 | 8.19 ± 3.33 | 1.39 ± 0.24 | 2.06 ± 0.13 |
| 3 | 525.02 ± 30.87 | 1803.97 ± 57.14 | 106.64 ± 6.64 | 327.34 ± 12.36 | 0.43 ± 0.01 | 5.72 ± 0.70 | 1.19 ± 0.03 | 1.78 ± 0.12 |
| 4 | 637.08 ± 42.84 | 2134.11 ± 145.82 | 134.36 ± 6.34 | 386.96 ± 28.23 | 0.58 ± 0.04 | 9.77 ± 1.46 | 1.44 ± 0.08 | 2.23 ± 0.22 |
| 5 | 537.10 ± 62.93 | 2028.48 ± 21.38 | 116.38 ± 4.68 | 375.43 ± 11.27 | 0.50 ± 0.02 | 8.30 ± 1.90 | 1.84 ± 0.10 | 2.47 ± 0.14 |
| 6 | 572.19 ± 17.71 | 1792.12 ± 159.96 | 123.46 ± 3.27 | 359.87 ± 11.84 | 0.46 ± 0.03 | 7.97 ± 2.44 | 1.28 ± 0.05 | 1.90 ± 0.13 |
| 7 | 530.38 ± 73.33 | 1957.07 ± 296.86 | 116.94 ± 9.97 | 352.58 ± 31.62 | 0.59 ± 0.25 | 4.95 ± 0.73 | 1.26 ± 0.15 | 2.10 ± 0.16 |
| 8 | 503.39 ± 67.39 | 1944.17 ± 113.53 | 106.83 ± 3.48 | 387.64 ± 23.65 | 0.41 ± 0.00 | 5.44 ± 0.98 | 0.94 ± 0.04 | 2.28 ± 0.29 |
| 9 | 570.22 ± 77.11 | 1959.03 ± 151.69 | 119.42 ± 16.62 | 420.38 ± 19.97 | 0.43 ± 0.03 | 6.14 ± 1.20 | 0.99 ± 0.21 | 2.65 ± 0.14 |
| 10 | 521.82 ± 23.22 | 1524.09 ± 202.22 | 96.37 ± 2.82 | 343.90 ± 6.17 | 0.40 ± 0.01 | 4.46 ± 0.24 | 1.16 ± 0.07 | 1.48 ± 0.11 |
| 11 | 536.48 ± 47.00 | 1420.33 ± 137.58 | 87.71 ± 6.40 | 333.49 ± 20.30 | 0.37 ± 0.04 | 3.82 ± 0.69 | 0.85 ± 0.17 | 1.20 ± 0.16 |
| 12 | 543.00 ± 73.70 | 2195.39 ± 64.11 | 120.07 ± 17.71 | 390.77 ± 40.80 | 0.25 ± 0.06 | 4.80 ± 0.84 | 0.98 ± 0.14 | 2.05 ± 0.32 |
| 13 | 777.19 ± 137.72 | 1835.02 ± 96.57 | 143.49 ± 7.81 | 408.49 ± 28.90 | 0.30 ± 0.03 | 5.41 ± 1.68 | 1.23 ± 0.26 | 2.22 ± 0.16 |
| 14 | 635.06 ± 14.37 | 1836.12 ± 90.64 | 123.41 ± 6.31 | 397.96 ± 26.80 | 0.28 ± 0.02 | 3.82 ± 0.24 | 1.01 ± 0.02 | 1.65 ± 0.07 |
| 15 | 657.54 ± 149.14 | 2161.58 ± 31.84 | 141.73 ± 23.29 | 388.14 ± 63.09 | 0.30 ± 0.06 | 5.93 ± 1.46 | 1.08 ± 0.08 | 2.21 ± 0.50 |
| 16 | 607.41 ± 18.33 | 2099.98 ± 194.02 | 125.67 ± 4.41 | 381.26 ± 28.17 | 0.34 ± 0.15 | 4.84 ± 0.46 | 0.95 ± 0.11 | 2.11 ± 0.18 |
| 17 | 497.83 ± 17.61 | 1508.75 ± 51.24 | 101.44 ± 1.68 | 360.00 ± 28.02 | 0.43 ± 0.02 | 4.76 ± 0.46 | 1.24 ± 0.13 | 1.46 ± 0.12 |
| 18 | 945.87 ± 328.79 | 1876.05 ± 224.20 | 148.13 ± 41.10 | 395.80 ± 13.15 | 0.75 ± 0.22 | 12.85 ± 4.86 | 2.17 ± 0.35 | 2.10 ± 0.43 |
| 19 | 1364.61 ± 106.71 | 2342.73 ± 109.80 | 177.31 ± 12.71 | 430.31 ± 23.46 | 1.16 ± 0.16 | 21.34 ± 7.05 | 2.66 ± 0.23 | 1.94 ± 0.06 |
| 20 | 459.54 ± 69.01 | 1994.19 ± 219.00 | 108.28 ± 3.47 | 319.00 ± 35.88 | 0.39 ± 0.04 | 5.13 ± 0.07 | 1.24 ± 0.06 | 2.02 ± 0.28 |
| 21 | 576.31 ± 55.23 | 1560.48 ± 95.11 | 97.57 ± 2.59 | 417.48 ± 17.97 | 0.40 ± 0.01 | 4.97 ± 0.47 | 0.98 ± 0.05 | 1.92 ± 0.24 |
| 22 | 426.00 ± 48.75 | 1776.67 ± 32.15 | 117.67 ± 4.04 | 424.00 ± 16.64 | 0.47 ± 0.03 | 3.89 ± 0.43 | 1.25 ± 0.10 | 2.91 ± 0.12 |
| 23 | 366.33 ± 23.03 | 1810.00 ± 75.50 | 103.67 ± 2.52 | 358.00 ± 18.19 | 0.41 ± 0.03 | 4.17 ± 0.26 | 1.34 ± 0.06 | 2.63 ± 0.14 |
| 24 | 432.67 ± 47.88 | 2006.67 ± 85.05 | 106.00 ± 3.61 | 433.33 ± 9.02 | 0.48 ± 0.05 | 4.51 ± 0.30 | 1.38 ± 0.04 | 3.42 ± 0.07 |
| 25 | 843.97 ± 130.81 | 2124.25 ± 98.19 | 133.22 ± 7.55 | 425.38 ± 29.74 | 0.65 ± 0.16 | 11.58 ± 1.76 | 1.65 ± 0.15 | 2.14 ± 0.20 |
| 26 | 782.39 ± 251.83 | 2281.98 ± 190.84 | 130.92 ± 24.90 | 412.05 ± 40.37 | 0.54 ± 0.07 | 9.00 ± 2.77 | 1.75 ± 0.28 | 2.36 ± 0.34 |
| 27 | 755.90 ± 248.64 | 1989.48 ± 102.98 | 125.97 ± 24.19 | 412.85 ± 12.92 | 1.09 ± 0.97 | 9.15 ± 1.99 | 1.61 ± 0.10 | 1.75 ± 0.15 |
| 28 | 411.58 ± 9.88 | 1880.22 ± 41.37 | 101.23 ± 6.86 | 379.64 ± 22.80 | 0.44 ± 0.05 | 5.35 ± 0.62 | 1.33 ± 0.07 | 1.96 ± 0.13 |
| 29 | 522.39 ± 53.52 | 1833.84 ± 256.16 | 92.25 ± 2.59 | 443.07 ± 42.51 | 0.43 ± 0.01 | 6.92 ± 1.53 | 1.49 ± 0.11 | 2.12 ± 0.18 |
| 30 | 941.73 ± 218.47 | 2128.14 ± 129.35 | 151.89 ± 26.20 | 429.98 ± 24.97 | 0.56 ± 0.06 | 12.15 ± 3.29 | 2.12 ± 0.52 | 2.37 ± 0.12 |
| 31 | 876.40 ± 102.55 | 2102.62 ± 128.73 | 156.41 ± 12.37 | 483.21 ± 19.27 | 0.59 ± 0.14 | 6.64 ± 0.42 | 1.34 ± 0.03 | 2.67 ± 0.28 |
| 32 | 791.51 ± 161.52 | 1837.68 ± 201.69 | 125.58 ± 4.51 | 466.02 ± 41.75 | 0.33 ± 0.03 | 5.59 ± 0.25 | 1.33 ± 0.11 | 2.30 ± 0.48 |
| 33 | 1237.70 ± 188.43 | 2314.79 ± 81.55 | 180.46 ± 16.94 | 502.14 ± 32.98 | 0.40 ± 0.02 | 13.68 ± 7.78 | 1.21 ± 0.11 | 2.52 ± 1.26 |
| 54 | 1190.57 ± 94.93 | 2306.85 ± 106.74 | 141.69 ± 10.46 | 329.06 ± 26.69 | 0.70 ± 0.48 | 11.16 ± 0.73 | 1.24 ± 0.15 | 1.86 ± 0.10 |
| 55 | 1027.28 ± 56.25 | 1768.00 ± 38.65 | 140.40 ± 6.69 | 313.94 ± 15.25 | 0.48 ± 0.13 | 9.32 ± 0.46 | 1.02 ± 0.11 | 1.39 ± 0.28 |

**Table A6.** Results of free amino acids in 35 Chinese cabbage.

| Materials Name | Asp mg/kg | Thr mg/kg | Ser mg/kg | Asn mg/kg | Glu mg/kg | Gln mg/kg |
|---|---|---|---|---|---|---|
| 1 | 112.47 ± 7.73 | 52.30 ± 10.67 | 157.97 ± 22.08 | 277.83 ± 44.90 | 268.27 ± 24.73 | 614.30 ± 319.12 |
| 2 | 95.47 ± 6.44 | 58.87 ± 1.34 | 129.63 ± 4.20 | 300.23 ± 16.61 | 304.17 ± 40.91 | 802.23 ± 351.67 |
| 3 | 44.60 ± 6.42 | 44.40 ± 2.31 | 78.27 ± 1.81 | 213.97 ± 1.57 | 128.47 ± 15.46 | 560.37 ± 274.27 |
| 4 | 87.27 ± 10.40 | 49.13 ± 8.24 | 83.83 ± 9.50 | 151.27 ± 12.65 | 137.33 ± 9.91 | 423.83 ± 194.77 |
| 5 | 73.33 ± 9.87 | 87.20 ± 19.00 | 178.57 ± 36.59 | 447.13 ± 85.91 | 209.57 ± 28.52 | 1576.80 ± 855.89 |
| 6 | 90.60 ± 4.53 | 59.50 ± 0.95 | 86.90 ± 2.46 | 295.60 ± 15.52 | 289.67 ± 11.60 | 741.57 ± 208.07 |
| 7 | 109.93 ± 4.57 | 147.17 ± 10.74 | 151.60 ± 19.28 | 1005.20 ± 161.82 | 402.20 ± 97.04 | 1461.17 ± 248.25 |
| 8 | 85.63 ± 3.86 | 116.40 ± 20.75 | 139.60 ± 21.01 | 517.67 ± 42.47 | 358.97 ± 59.61 | 2100.13 ± 972.27 |
| 9 | 95.50 ± 18.93 | 115.80 ± 22.88 | 141.03 ± 19.56 | 544.53 ± 115.34 | 278.53 ± 94.63 | 2439.67 ± 600.22 |
| 10 | 72.90 ± 11.39 | 72.20 ± 11.95 | 140.87 ± 10.29 | 317.17 ± 46.59 | 150.57 ± 19.77 | 1489.13 ± 246.59 |
| 11 | 69.13 ± 1.87 | 63.27 ± 4.39 | 116.93 ± 23.65 | 290.30 ± 24.55 | 320.27 ± 43.50 | 1136.87 ± 174.68 |
| 12 | 100.57 ± 20.13 | 86.13 ± 52.27 | 117.33 ± 22.09 | 503.53 ± 455.87 | 381.77 ± 58.50 | 1180.00 ± 110.33 |
| 13 | 89.97 ± 2.18 | 106.13 ± 8.35 | 139.13 ± 7.86 | 515.33 ± 36.29 | 434.10 ± 20.80 | 1656.60 ± 267.49 |
| 14 | 79.83 ± 7.05 | 55.97 ± 1.65 | 131.20 ± 16.53 | 318.93 ± 31.37 | 249.83 ± 10.05 | 1272.73 ± 211.27 |

**Table A6.** *Cont.*

| Materials Name | Asp mg/kg | Thr mg/kg | Ser mg/kg | Asn mg/kg | Glu mg/kg | Gln mg/kg |
|---|---|---|---|---|---|---|
| 15 | 90.00 ± 7.03 | 100.10 ± 33.20 | 128.80 ± 28.18 | 604.33 ± 394.23 | 332.03 ± 24.23 | 1448.73 ± 754.08 |
| 16 | 83.87 ± 8.33 | 68.70 ± 9.08 | 99.10 ± 7.84 | 314.23 ± 30.06 | 291.13 ± 15.79 | 1134.40 ± 187.84 |
| 17 | 61.17 ± 9.77 | 43.30 ± 12.16 | 96.57 ± 17.00 | 240.13 ± 50.78 | 207.30 ± 10.85 | 612.60 ± 320.16 |
| 18 | 83.93 ± 16.78 | 67.67 ± 10.84 | 215.03 ± 29.24 | 294.67 ± 49.12 | 291.03 ± 17.50 | 862.97 ± 204.12 |
| 19 | 94.53 ± 15.67 | 65.57 ± 4.65 | 182.60 ± 24.56 | 242.53 ± 16.40 | 250.50 ± 19.22 | 790.53 ± 101.86 |
| 20 | 132.10 ± 19.77 | 140.97 ± 15.51 | 146.60 ± 8.34 | 404.63 ± 57.05 | 417.10 ± 61.02 | 1258.53 ± 286.36 |
| 21 | 85.37 ± 4.11 | 71.33 ± 13.17 | 179.57 ± 24.52 | 336.50 ± 68.37 | 123.70 ± 6.22 | 949.53 ± 205.80 |
| 22 | 28.63 ± 10.84 | 122.87 ± 29.61 | 183.63 ± 49.00 | 385.87 ± 80.52 | 113.30 ± 17.61 | 2406.47 ± 547.15 |
| 23 | 35.50 ± 1.49 | 127.03 ± 47.08 | 181.57 ± 35.96 | 435.63 ± 105.84 | 88.17 ± 25.08 | 2480.57 ± 497.30 |
| 24 | 90.53 ± 22.45 | 165.43 ± 33.83 | 264.73 ± 37.07 | 520.97 ± 74.28 | 231.53 ± 16.65 | 2702.27 ± 529.12 |
| 25 | 79.00 ± 8.25 | 40.30 ± 5.66 | 63.43 ± 14.09 | 94.33 ± 41.17 | 81.43 ± 29.41 | 249.97 ± 172.92 |
| 26 | 109.40 ± 7.45 | 66.53 ± 8.95 | 110.43 ± 16.43 | 165.97 ± 43.81 | 174.70 ± 36.70 | 555.70 ± 182.63 |
| 27 | 94.97 ± 12.49 | 51.13 ± 2.21 | 120.33 ± 20.65 | 186.07 ± 35.10 | 175.33 ± 50.75 | 512.23 ± 157.55 |
| 28 | 63.60 ± 13.81 | 51.70 ± 4.50 | 149.63 ± 13.84 | 193.03 ± 13.11 | 294.97 ± 16.66 | 513.97 ± 56.09 |
| 29 | 75.07 ± 12.76 | 52.40 ± 4.40 | 148.10 ± 10.35 | 214.77 ± 14.93 | 252.60 ± 20.13 | 577.33 ± 111.00 |
| 30 | 132.53 ± 14.06 | 72.10 ± 6.71 | 145.17 ± 32.32 | 143.60 ± 26.52 | 188.57 ± 20.57 | 501.97 ± 114.65 |
| 31 | 119.63 ± 8.27 | 79.50 ± 9.04 | 142.20 ± 11.26 | 350.30 ± 60.77 | 251.83 ± 11.05 | 1675.67 ± 313.64 |
| 32 | 67.70 ± 7.84 | 52.00 ± 10.77 | 100.73 ± 14.70 | 260.57 ± 50.15 | 209.27 ± 25.65 | 1282.50 ± 472.54 |
| 33 | 141.03 ± 18.08 | 72.40 ± 7.69 | 252.63 ± 25.10 | 135.07 ± 73.63 | 143.27 ± 14.23 | 638.53 ± 308.68 |
| 34 | 105.07 ± 1.63 | 32.77 ± 0.59 | 87.83 ± 1.65 | 85.77 ± 4.86 | 49.13 ± 1.15 | 314.73 ± 13.32 |
| 35 | 112.17 ± 4.86 | 43.70 ± 1.90 | 73.23 ± 3.06 | 60.40 ± 8.08 | 55.47 ± 8.33 | 213.53 ± 5.91 |

| Materials Name | Pro mg/kg | Gly mg/kg | Ala mg/kg | Val mg/kg | Cys mg/kg | Met |
|---|---|---|---|---|---|---|
| 1 | 31.87 ± 6.18 | 22.67 ± 2.52 | 163.33 ± 26.08 | 63.40 ± 16.96 | 8.63 ± 1.80 | mg/kg |
| 2 | 33.90 ± 1.39 | 35.73 ± 2.15 | 285.87 ± 11.63 | 76.40 ± 11.44 | 21.17 ± 3.10 | 10.97 ± 2.15 |
| 3 | 19.87 ± 1.25 | 28.30 ± 2.72 | 137.57 ± 12.40 | 53.23 ± 8.35 | 13.77 ± 2.57 | 15.27 ± 2.45 |
| 4 | 58.90 ± 19.51 | 23.70 ± 2.21 | 155.37 ± 15.30 | 54.57 ± 11.46 | 31.87 ± 1.14 | 9.90 ± 2.46 |
| 5 | 23.00 ± 3.27 | 50.43 ± 8.96 | 272.93 ± 59.59 | 88.27 ± 25.18 | 18.40 ± 1.87 | 10.63 ± 1.86 |
| 6 | 40.33 ± 6.93 | 26.27 ± 0.67 | 124.17 ± 3.82 | 67.77 ± 6.39 | 17.07 ± 2.54 | 20.13 ± 1.47 |
| 7 | 57.30 ± 38.29 | 63.40 ± 5.77 | 410.97 ± 28.07 | 115.17 ± 7.23 | 38.03 ± 13.64 | 11.23 ± 5.33 |
| 8 | 47.87 ± 7.45 | 67.73 ± 14.05 | 546.67 ± 138.05 | 98.57 ± 4.06 | 31.87 ± 6.87 | 15.33 ± 7.98 |
| 9 | 50.77 ± 11.39 | 63.73 ± 9.53 | 510.73 ± 83.99 | 91.13 ± 14.23 | 32.53 ± 5.56 | 18.60 ± 5.96 |
| 10 | 88.70 ± 14.16 | 34.10 ± 2.38 | 351.17 ± 34.10 | 72.30 ± 18.89 | 11.57 ± 3.10 | 20.67 ± 2.97 |
| 11 | 52.03 ± 14.82 | 30.07 ± 4.04 | 160.30 ± 28.95 | 72.77 ± 9.71 | 14.20 ± 2.23 | 15.53 ± 1.85 |
| 12 | 66.13 ± 29.12 | 47.87 ± 4.53 | 392.93 ± 20.80 | 85.10 ± 7.36 | 28.10 ± 2.36 | 15.93 ± 0.71 |
| 13 | 68.83 ± 13.87 | 47.60 ± 2.17 | 379.57 ± 35.07 | 102.30 ± 17.71 | 37.67 ± 9.25 | 17.70 ± 1.65 |
| 14 | 94.33 ± 5.89 | 29.87 ± 1.93 | 246.53 ± 16.51 | 68.30 ± 16.49 | 15.03 ± 2.86 | 18.67 ± 2.06 |
| 15 | 76.73 ± 14.72 | 49.70 ± 12.70 | 407.03 ± 43.30 | 102.77 ± 39.18 | 31.47 ± 12.96 | 16.07 ± 1.59 |
| 16 | 53.07 ± 1.61 | 44.70 ± 2.95 | 290.50 ± 36.68 | 77.77 ± 20.53 | 22.83 ± 3.69 | 20.83 ± 5.25 |
| 17 | 20.97 ± 2.33 | 25.73 ± 4.32 | 167.83 ± 37.02 | 56.33 ± 16.26 | 10.00 ± 1.22 | 16.03 ± 2.29 |
| 18 | 74.10 ± 8.53 | 27.33 ± 1.81 | 199.33 ± 15.51 | 70.13 ± 8.09 | 17.50 ± 2.87 | 11.57 ± 0.95 |
| 19 | 107.77 ± 14.87 | 23.20 ± 2.44 | 144.80 ± 17.97 | 73.83 ± 6.13 | 11.17 ± 1.10 | 13.90 ± 3.59 |
| 20 | 56.60 ± 18.36 | 30.90 ± 7.54 | 220.20 ± 26.20 | 89.97 ± 11.04 | 13.97 ± 3.52 | 20.67 ± 5.14 |
| 21 | 43.23 ± 2.60 | 29.00 ± 3.10 | 177.10 ± 27.27 | 71.60 ± 16.46 | 13.67 ± 3.09 | 19.40 ± 5.60 |
| 22 | 59.97 ± 17.68 | 57.37 ± 9.76 | 278.63 ± 68.74 | 86.30 ± 14.86 | 37.60 ± 4.57 | 17.93 ± 2.50 |
| 23 | 47.83 ± 6.07 | 55.07 ± 8.20 | 185.87 ± 39.44 | 94.10 ± 18.27 | 36.87 ± 7.85 | 14.27 ± 2.58 |
| 24 | 193.23 ± 42.06 | 62.67 ± 9.60 | 471.53 ± 86.63 | 152.97 ± 28.42 | 43.47 ± 9.07 | 16.53 ± 2.41 |
| 25 | 68.20 ± 10.08 | 20.27 ± 6.85 | 112.33 ± 32.81 | 47.47 ± 0.78 | 17.77 ± 6.82 | 21.47 ± 1.70 |
| 26 | 158.50 ± 75.40 | 26.17 ± 5.25 | 161.50 ± 31.28 | 69.43 ± 21.68 | 19.43 ± 4.65 | 13.20 ± 4.76 |
| 27 | 87.37 ± 10.95 | 24.10 ± 6.55 | 155.93 ± 39.14 | 61.37 ± 5.36 | 16.37 ± 6.60 | 17.20 ± 3.11 |
| 28 | 66.33 ± 11.15 | 37.60 ± 7.62 | 244.70 ± 26.85 | 68.17 ± 15.10 | 32.50 ± 8.34 | 14.23 ± 5.66 |
| 29 | 79.87 ± 2.50 | 33.37 ± 2.63 | 192.73 ± 3.45 | 72.53 ± 18.66 | 24.30 ± 4.18 | 15.97 ± 4.05 |
| 30 | 266.67 ± 35.51 | 25.43 ± 6.21 | 319.93 ± 89.40 | 93.00 ± 2.26 | 11.72 ± 1.88 | 16.47 ± 4.30 |
| 31 | 144.07 ± 24.17 | 26.37 ± 3.64 | 254.53 ± 32.16 | 91.07 ± 7.58 | 14.13 ± 2.27 | 29.10 ± 0.46 |
| 32 | 122.10 ± 51.69 | 30.73 ± 7.43 | 242.30 ± 58.57 | 78.73 ± 25.48 | 13.11 ± 3.82 | 23.47 ± 2.30 |
| 33 | 136.23 ± 42.32 | 24.67 ± 3.98 | 228.33 ± 50.16 | 82.83 ± 18.56 | 10.43 ± 3.00 | 18.77 ± 2.97 |
| 34 | 94.43 ± 1.19 | 13.70 ± 0.17 | 91.97 ± 1.79 | 38.43 ± 2.26 | 2.92 ± 0.16 | 30.17 ± 4.35 |
| 35 | 48.27 ± 0.67 | 22.83 ± 1.72 | 112.93 ± 5.28 | 40.70 ± 4.16 | 6.81 ± 1.26 | 18.17 ± 0.46 |

| Materials Name | Ile mg/kg | Leu mg/kg | Tyr mg/kg | Phe mg/kg | γ-Aba mg/kg | His mg/kg |
|---|---|---|---|---|---|---|
| 1 | 32.37 ± 6.77 | 36.60 ± 3.26 | 22.63 ± 3.65 | 29.50 ± 5.35 | 46.43 ± 2.48 | 23.57 ± 4.01 |
| 2 | 37.37 ± 1.19 | 47.83 ± 4.94 | 23.57 ± 1.56 | 32.83 ± 5.89 | 51.83 ± 5.05 | 26.17 ± 2.90 |
| 3 | 25.63 ± 1.66 | 27.37 ± 2.45 | 13.83 ± 1.47 | 17.73 ± 2.76 | 108.73 ± 12.37 | 16.07 ± 0.86 |
| 4 | 24.20 ± 3.82 | 30.90 ± 3.47 | 14.27 ± 1.31 | 18.93 ± 3.60 | 206.83 ± 21.25 | 12.13 ± 1.65 |
| 5 | 51.23 ± 10.12 | 39.60 ± 4.06 | 18.37 ± 2.08 | 25.00 ± 4.79 | 170.20 ± 25.24 | 39.60 ± 7.65 |
| 6 | 34.13 ± 2.61 | 39.47 ± 6.77 | 25.27 ± 3.95 | 32.00 ± 3.47 | 55.20 ± 9.44 | 25.67 ± 1.15 |
| 7 | 69.77 ± 6.94 | 49.13 ± 9.46 | 26.43 ± 5.35 | 43.73 ± 1.72 | 188.57 ± 87.05 | 60.57 ± 9.00 |
| 8 | 78.80 ± 14.35 | 53.30 ± 7.99 | 27.67 ± 4.98 | 45.40 ± 9.37 | 200.13 ± 24.35 | 46.43 ± 3.56 |
| 9 | 72.73 ± 15.42 | 55.90 ± 9.77 | 28.07 ± 4.32 | 48.57 ± 12.27 | 205.80 ± 34.57 | 45.13 ± 9.96 |
| 10 | 41.00 ± 4.92 | 40.60 ± 3.97 | 25.50 ± 2.34 | 26.60 ± 4.16 | 199.40 ± 11.51 | 29.77 ± 4.35 |
| 11 | 43.63 ± 1.47 | 48.60 ± 0.52 | 25.07 ± 1.55 | 32.60 ± 2.29 | 22.07 ± 1.70 | 26.37 ± 0.47 |
| 12 | 50.90 ± 13.71 | 46.40 ± 3.25 | 23.30 ± 1.76 | 37.53 ± 10.89 | 46.20 ± 22.45 | 30.63 ± 17.38 |
| 13 | 60.00 ± 7.81 | 52.73 ± 11.85 | 25.87 ± 3.07 | 38.47 ± 1.27 | 23.57 ± 9.45 | 41.07 ± 2.76 |
| 14 | 37.90 ± 4.10 | 44.20 ± 4.67 | 23.30 ± 1.71 | 28.00 ± 1.90 | 142.80 ± 17.75 | 25.53 ± 1.44 |
| 15 | 60.40 ± 16.63 | 55.80 ± 15.49 | 27.60 ± 4.64 | 38.23 ± 7.77 | 87.77 ± 60.49 | 36.57 ± 15.30 |
| 16 | 43.33 ± 3.87 | 43.80 ± 7.33 | 20.93 ± 2.40 | 27.13 ± 4.22 | 107.80 ± 17.82 | 23.43 ± 0.81 |
| 17 | 30.20 ± 7.28 | 32.57 ± 2.27 | 18.83 ± 3.49 | 23.83 ± 6.44 | 23.40 ± 3.48 | 17.90 ± 2.26 |
| 18 | 33.33 ± 1.50 | 44.80 ± 3.22 | 23.20 ± 2.69 | 30.83 ± 3.48 | 80.90 ± 3.76 | 27.33 ± 4.46 |
| 19 | 40.43 ± 2.68 | 61.20 ± 7.71 | 29.77 ± 3.98 | 38.63 ± 3.19 | 133.73 ± 24.16 | 28.50 ± 1.92 |
| 20 | 62.77 ± 13.48 | 55.30 ± 14.25 | 31.37 ± 8.05 | 41.50 ± 12.64 | 58.50 ± 11.26 | 34.33 ± 3.35 |
| 21 | 45.00 ± 7.71 | 45.67 ± 6.72 | 27.73 ± 4.58 | 42.70 ± 12.50 | 144.60 ± 8.20 | 30.23 ± 6.69 |
| 22 | 43.57 ± 12.61 | 33.37 ± 5.30 | 27.13 ± 2.02 | 19.47 ± 2.63 | 154.03 ± 41.45 | 34.87 ± 10.04 |
| 23 | 50.63 ± 13.92 | 40.27 ± 5.29 | 31.17 ± 3.04 | 25.93 ± 3.87 | 150.93 ± 9.03 | 49.23 ± 13.88 |
| 24 | 92.70 ± 19.63 | 55.57 ± 2.73 | 36.20 ± 0.85 | 31.40 ± 2.78 | 205.77 ± 28.22 | 59.30 ± 14.42 |
| 25 | 24.83 ± 5.87 | 44.73 ± 7.16 | 21.93 ± 3.61 | 28.20 ± 3.22 | 181.07 ± 23.26 | 10.53 ± 2.31 |
| 26 | 34.57 ± 5.52 | 57.83 ± 16.61 | 26.60 ± 6.41 | 33.27 ± 9.17 | 217.87 ± 37.31 | 15.23 ± 2.27 |
| 27 | 31.47 ± 3.81 | 48.73 ± 6.20 | 25.07 ± 2.63 | 31.73 ± 1.58 | 165.10 ± 35.96 | 16.43 ± 1.76 |
| 28 | 36.13 ± 4.61 | 50.10 ± 3.56 | 25.80 ± 2.76 | 31.37 ± 3.73 | 110.00 ± 0.98 | 15.77 ± 0.64 |
| 29 | 38.73 ± 2.30 | 56.63 ± 3.30 | 29.57 ± 1.15 | 34.57 ± 2.11 | 155.90 ± 13.20 | 18.20 ± 1.31 |
| 30 | 67.57 ± 6.38 | 81.53 ± 1.45 | 44.13 ± 2.31 | 53.23 ± 10.04 | 349.93 ± 34.92 | 21.17 ± 1.53 |
| 31 | 69.00 ± 5.91 | 73.80 ± 5.73 | 39.20 ± 2.43 | 46.27 ± 7.55 | 262.40 ± 11.00 | 40.33 ± 5.51 |
| 32 | 47.50 ± 11.10 | 47.23 ± 7.35 | 22.23 ± 3.17 | 23.83 ± 5.09 | 161.70 ± 15.60 | 22.90 ± 5.21 |
| 33 | 40.67 ± 7.09 | 73.60 ± 9.93 | 32.93 ± 1.89 | 37.00 ± 3.42 | 357.50 ± 15.78 | 16.63 ± 2.59 |
| 34 | 23.63 ± 0.42 | 43.40 ± 0.46 | 18.83 ± 0.15 | 22.33 ± 4.65 | 214.63 ± 3.98 | 8.81 ± 0.09 |
| 35 | 24.93 ± 3.20 | 52.37 ± 6.01 | 23.87 ± 3.02 | 27.30 ± 4.03 | 209.60 ± 8.70 | 10.00 ± 0.46 |

| Materials Name | Trp mg/kg | Orn mg/kg | Lys mg/kg | Arg mg/kg |
|---|---|---|---|---|
| 1 | 1.71 ± 0.69 | 22.67 ± 4.51 | 48.50 ± 4.04 | 248.00 ± 45.56 |
| 2 | 2.54 ± 0.89 | 13.10 ± 1.48 | 57.07 ± 6.86 | 197.20 ± 19.66 |
| 3 | 1.68 ± 0.58 | 15.50 ± 1.21 | 34.90 ± 2.51 | 150.97 ± 8.46 |
| 4 | 1.58 ± 0.71 | 15.53 ± 1.24 | 37.67 ± 4.04 | 75.00 ± 14.31 |
| 5 | 1.98 ± 0.52 | 40.47 ± 8.81 | 46.27 ± 5.51 | 245.57 ± 34.99 |
| 6 | 3.72 ± 3.31 | 18.30 ± 0.26 | 51.87 ± 7.01 | 114.97 ± 7.11 |
| 7 | 5.64 ± 6.22 | 45.63 ± 1.23 | 66.80 ± 13.20 | 316.33 ± 58.31 |
| 8 | 6.17 ± 4.55 | 27.43 ± 3.32 | 70.60 ± 10.63 | 305.00 ± 16.36 |
| 9 | 4.74 ± 1.70 | 28.57 ± 7.06 | 72.47 ± 9.09 | 332.03 ± 80.39 |
| 10 | 2.32 ± 2.37 | 26.63 ± 2.38 | 52.17 ± 6.07 | 138.67 ± 24.69 |
| 11 | 2.26 ± 1.75 | 17.27 ± 0.49 | 66.17 ± 2.64 | 247.53 ± 2.94 |
| 12 | 2.91 ± 2.74 | 27.07 ± 11.04 | 62.20 ± 4.16 | 225.93 ± 154.65 |
| 13 | 3.36 ± 2.79 | 24.53 ± 2.02 | 68.97 ± 15.29 | 310.43 ± 10.72 |
| 14 | 2.82 ± 3.14 | 22.20 ± 1.28 | 57.93 ± 6.20 | 162.63 ± 11.65 |
| 15 | 2.72 ± 1.02 | 28.83 ± 9.62 | 73.33 ± 19.29 | 277.63 ± 126.87 |
| 16 | 2.10 ± 0.90 | 20.37 ± 0.95 | 59.07 ± 9.15 | 199.73 ± 18.61 |
| 17 | 1.54 ± 0.45 | 10.63 ± 2.87 | 42.73 ± 2.87 | 173.57 ± 41.38 |
| 18 | 4.85 ± 4.53 | 13.63 ± 1.50 | 62.67 ± 7.78 | 190.27 ± 40.20 |
| 19 | 6.37 ± 5.60 | 14.93 ± 1.36 | 78.67 ± 11.04 | 180.67 ± 14.97 |
| 20 | 5.17 ± 4.55 | 23.03 ± 3.59 | 61.23 ± 16.92 | 226.87 ± 30.12 |
| 21 | 3.83 ± 1.29 | 36.37 ± 5.97 | 43.53 ± 37.13 | 185.77 ± 34.63 |
| 22 | 4.46 ± 0.64 | 18.13 ± 7.00 | 51.07 ± 8.37 | 287.20 ± 70.54 |
| 23 | 5.61 ± 0.58 | 20.93 ± 4.90 | 57.30 ± 11.40 | 191.60 ± 55.01 |
| 24 | 5.03 ± 3.20 | 35.90 ± 6.85 | 71.77 ± 4.65 | 451.73 ± 65.35 |
| 25 | 4.90 ± 4.61 | 9.70 ± 2.46 | 57.97 ± 11.29 | 65.90 ± 19.07 |
| 26 | 4.86 ± 3.62 | 22.90 ± 4.13 | 71.43 ± 16.29 | 97.63 ± 10.89 |
| 27 | 4.07 ± 3.44 | 12.73 ± 2.00 | 65.73 ± 8.57 | 146.93 ± 31.30 |
| 28 | 2.24 ± 0.52 | 14.73 ± 0.51 | 65.60 ± 7.07 | 125.13 ± 17.74 |
| 29 | 2.37 ± 0.27 | 14.33 ± 2.25 | 74.53 ± 4.68 | 158.27 ± 35.51 |
| 30 | 7.04 ± 4.85 | 28.27 ± 3.74 | 109.47 ± 9.04 | 90.07 ± 11.84 |
| 31 | 6.01 ± 4.34 | 28.20 ± 2.50 | 102.10 ± 2.95 | 293.40 ± 23.03 |
| 32 | 2.61 ± 1.89 | 23.43 ± 4.67 | 60.07 ± 8.37 | 143.63 ± 26.06 |
| 33 | 4.63 ± 3.83 | 9.63 ± 0.95 | 100.17 ± 12.42 | 128.70 ± 19.64 |
| 34 | 3.32 ± 4.51 | 5.17 ± 0.29 | 47.77 ± 2.37 | 55.07 ± 2.84 |
| 35 | 3.37 ± 4.59 | 4.13 ± 0.43 | 64.70 ± 4.83 | 58.60 ± 4.10 |

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
