# Peer review of "Comprehensive Evaluation of Nutritional Qualities of Chinese Cabbage (Brassica rapa ssp. pekinensis) Varieties Based on Multivariate Statistical Analysis"

_horticulturae, doi:10.3390/horticulturae9121264_

Round 1
Reviewer 1 Report
Comments and Suggestions for Authors
Comments in attached file.

Reviewer 2 Report
Comments and Suggestions for Authors
Overall, this manuscript is well written and developed.
In the discussion section, lines 423-425, please explain further the reason for these differences (high coefficient of variation), especially within the same sample (triplicate).
Reviewer 3 Report
Comments and Suggestions for Authors
Scientific Review Form
In this paper, the authors present an analysis of cabbage varieties that were collected in China. The statistical analysis is well-made, and the manuscript is well-written. The manuscript is suitable for Horticulturae because it is in the journal's aim and scope. The author's findings are significant because they demonstrate that the nutritional quality of Chinese cabbage is different. However, the manuscript needs some polishing before publication in a high-standard journal such as Horticulturae. In particular, the material and methods section must be improved to be reproducible.
INTRODUCTION
L40-L42: "According to…China". There is a reference?
L49-L52: "Commodity..components". Is it true that the smell and the taste of Chinese cabbage differ from the other? Do you have any references?
L67: Add space before [9]. If there is a possibility, use a more recent reference.
L67-L71: I suggest rephrasing in short periods and avoiding repeating the same words in the same way. Honestly, this seems AI-generated.
L88-L91: Why is HCA the acronym for different things?
METHODS
There are no sufficient data about the analytical method that you use. Repeat. Say "was carried out [32]" in L137 is not reproducible. The same with the other analytical methods. What about analytical parameters? LOD? LOQ? Repeteability? Accuracy? Recoveries? The methods need internal standards. Did you buy them? Where? The sample preparation needs more data or the paper is not reproducible. I request a chromatogram from the ICP-OES. Explain how you perform the analysis in the supplementary materials, not just with the references.
L154-L155: I can not enter the site. What you did do with SPSS and what with the Online website?
RESULTS
L212-L223: As a result, you don’t have to discuss results.
Figure 2 is low resolution.
L291: Add space after “including”
L302-L303: Add the caption about MA, CA, OA, VC, TAA etc. Remove “(“ from K.
Comments on the Quality of English LanguageThe quality of the English Language is ok.
Reviewer 4 Report
Comments and Suggestions for Authors
In this manuscript, a multivariate analysis of 35 species of Cabbage (Brassica rapa ssp. pekinensis) is proposed to evaluate their nutritional quality. It is a novel and relevant study that correctly applies statistical tools of analysis. In general, it is a conclusive and rigorous study, which is up to the standards of a scientific publication in the field. Although multivariate analyses of Chinese Cabbage are available for metabolic profiling, this study provides an interesting view from a compositional approach.
In the introduction, I see very long paragraphs. An adjustment is needed.
In the methodology, the determination of sugar components by ion chromatography should explain in more detail the operational conditions and the equipment used to perform the quantification.
In the study, and specifically in the analysis of results, the reason for taking 35 types of Chinese cabbage is not clear. Please explain.
The conclusions should be reformulated since they do not specifically mention the results obtained. A general overview of the results is given, but it could be made more explicit in view of the quality of the results obtained.
Round 2
Reviewer 3 Report
Comments and Suggestions for Authors
The authors answered all my comments.